# Lifelong Policy Gradient Learning
# of Factored Policies
# for Faster Training Without Forgetting

**Jorge A. Mendez**[1], **Boyu Wang**[2], and **Eric Eaton**[1]

[1]Dept. of Computer and Information Science
University of Pennsylvania
{mendezme,eeaton}@seas.upenn.edu

[2]Dept. of Computer Science
University of Western Ontario
bwang@csd.uwo.ca

## Abstract

Policy gradient methods have shown success in learning control policies for high-dimensional dynamical systems. Their biggest downside is the amount of exploration they require before yielding high-performing policies. In a lifelong learning setting, in which an agent is faced with multiple consecutive tasks over its lifetime, reusing information from previously seen tasks can substantially accelerate the learning of new tasks. We provide a novel method for lifelong policy gradient learning that trains lifelong function approximators directly via policy gradients, allowing the agent to benefit from accumulated knowledge throughout the entire training process. We show empirically that our algorithm learns faster and converges to better policies than single-task and lifelong learning baselines, and completely avoids catastrophic forgetting on a variety of challenging domains.

## 1 Introduction

Policy gradient (PG) methods have been successful in learning control policies on high-dimensional, continuous systems [33, 23, 34]. However, like most methods for reinforcement learning (RL), they require the agent to interact with the world extensively before outputting a functional policy. In some settings, this experience is prohibitively expensive, such as when training an actual physical system.

If an agent is expected to learn multiple consecutive tasks over its lifetime, then we would want it to leverage knowledge from previous tasks to accelerate the learning of new tasks. This is the premise of lifelong RL methods. Most previous work in this field has considered the existence of a central policy that can be used to solve all tasks the agent will encounter [21, 35]. If the tasks are sufficiently related, this model serves as a good starting point for learning new tasks, and the main problem becomes how to avoid forgetting the knowledge required to solve tasks encountered early in the agent's lifetime.

However, in many cases, the tasks the agent will encounter are less closely related, and so a single policy is insufficient for solving all tasks. A typical approach for handling this (more realistic) setting is to train a separate policy for each new task, and then use information obtained during training to find commonalities to previously seen tasks and use these relations to improve the learned policy [6, 5, 19]. Note that this only enables the agent to improve policy performance *after* an initial policy has been trained. Such methods have been successful in outperforming the original policies trained independently for each task, but unfortunately do not allow the agent to reuse knowledge from previous tasks to more efficiently search for policies, and so the learning itself is not accelerated.

We propose a novel framework for lifelong RL via PG learning that automatically leverages prior experience *during* the training process of each task. In order to enable learning highly diverse tasks, we follow prior work in lifelong RL by searching over factored representations of the policy-

parameter space to learn both a shared repository of knowledge and a series of task-specific mappings to constitute individual task policies from the shared knowledge.

Our algorithm, *lifelong PG: faster training without forgetting* (LPG-FTW) yields high-performing policies on a variety of benchmark problems with less experience than independently learning each task, and avoids the problem of *catastrophic forgetting* [24]. Moreover, we show theoretically that LPG-FTW is guaranteed to converge to a particular approximation to the multi-task objective.

## 2 Related work

A large body of work in lifelong RL is based on parameter sharing, where the underlying relations among multiple tasks are captured by the model parameters. The key problem is designing how to share parameters across tasks in such a way that subsequent tasks benefit from the earlier tasks, and the modification of the parameters by future tasks does not hinder performance of the earlier tasks.

Two broad categories of methods have arisen that differ in the way they share parameters.

The first class of lifelong RL techniques, which we will call single-model, assumes that there is one model that works for all tasks. These algorithms follow some standard single-task learning (STL) PG algorithm, but modify the PG objective to encourage transfer across tasks. A prominent example is elastic weight consolidation (EWC) [21], which imposes a quadratic penalty for deviating from earlier tasks' parameters to avoid forgetting. This idea has been extended by modifying the exact form of the penalty [22, 42, 26, 29, 10, 37], but most of these approaches have not been evaluated in RL. An alternative approach has also been proposed that first trains an auxiliary model for each new task and subsequently discards it and distills any new knowledge into the single shared model [35]. In order for single-model methods to work, they require one of two assumptions to hold: either all tasks must be very similar, or the model must be over-parameterized to capture variations among tasks. The first assumption is clearly quite restrictive, as it would preclude the agent from learning highly varied tasks. The second, we argue, is just as restrictive, since the over-parameterization is finite, and so the model can become saturated after a (typically small) number of tasks.

The second class, which we will call multi-model, assumes that there is a set of shared parameters, representing a collection of models, and a set of task-specific parameters, to select a combination of these models for the current task. A classical example is PG-ELLA [6, 5, 19], which assumes that each task's parameters are factored as a linear combination of dictionary elements. A similar approach uses tensor factorization for deep nets [43], but in a batch multi-task setting. In a first stage, these methods learn a policy for each task in isolation (i.e., ignoring any information from other tasks) to determine similarity to previous policies, and in a second stage, the parameters are factored to improve performance via transfer. The downside of this is that the agent does not benefit from prior experience during initial training, which is critical for efficient learning in a lifelong RL setting.

Our approach, LPG-FTW, uses multiple models like the latter category, but learns these models directly via PG learning like the former class. This enables LPG-FTW to be flexible and handle highly varied tasks while also benefiting from prior information during the learning process, thus accelerating the training. A similar approach has been explored in the context of model-based RL [25], but their focus was discovering when new tasks were encountered in the absence of task indicators.

Other approaches store experiences in memory for future replay [18, 30] or use a separate model for each task [31, 14]. The former is not applicable to PG methods without complex and often unreliable importance sampling techniques, while the latter is infeasible when the number of tasks grows large.

Meta-RL [9, 11, 15, 8] and multi-task RL [27, 36, 40, 43] also seek to accelerate learning by reusing information from different tasks, but in those settings the agent does not handle tasks arriving sequentially and the consequent problem of catastrophic forgetting. Instead, there is a large batch of tasks available for training and evaluation is done either on the same batch or on a target task.

## 3 Reinforcement learning via policy gradients

In a Markov decision process (MDP) $\langle \mathcal{X}, \mathcal{U}, T, R, \gamma \rangle$, $\mathcal{X} \subseteq \mathbb{R}^d$ is the set of states, $\mathcal{U} \subseteq \mathbb{R}^m$ is the set of actions, $T : \mathcal{X} \times \mathcal{U} \times \mathcal{X} \mapsto [0, 1]$ is the probability $P(\boldsymbol{x}' \mid \boldsymbol{x}, \boldsymbol{u})$ of going to state $\boldsymbol{x}'$ after executing action $\boldsymbol{u}$ in state $\boldsymbol{x}$, $R : \mathcal{X} \times \mathcal{U} \mapsto \mathbb{R}$ is the reward function measuring the goodness of a state-action

pair, and $\gamma \in [0, 1)$ is the discount factor for future rewards. A policy $\pi : \mathcal{X} \times \mathcal{U} \mapsto [0, 1]$ prescribes the agent's behavior as a probability $P(\boldsymbol{u} \mid \boldsymbol{x})$ of selecting action $\boldsymbol{u}$ in state $\boldsymbol{x}$. The goal of RL is to find the policy $\pi^*$ that maximizes the expected returns $\mathbb{E}\left[\sum_{i=0}^{\infty} \gamma^i R_i\right]$, where $R_i = R(\boldsymbol{x}_i, \boldsymbol{u}_i)$.

PG algorithms have shown success in solving continuous RL problems by assuming that the policy $\pi_{\boldsymbol{\theta}}$ is parameterized by $\boldsymbol{\theta} \in \mathbb{R}^d$ and searching for the set of parameters $\boldsymbol{\theta}^*$ that optimizes the long-term rewards: $\mathcal{J}(\boldsymbol{\theta}) = \mathbb{E}\left[\sum_{i=0}^{\infty} \gamma^i R_i\right]$ [33, 23, 34]. Different approaches use varied strategies for estimating the gradient $\nabla_{\boldsymbol{\theta}} \mathcal{J}(\boldsymbol{\theta})$. However, the common high-level idea is to use the current policy $\pi_{\boldsymbol{\theta}}$ to sample trajectories of interaction with the environment, and then estimating the gradient as the average of some function of the features and rewards encountered through the trajectories.

# 4   The lifelong learning problem

We frame lifelong PG learning as online multi-task learning of policy parameters. The agent will face a sequence of tasks $\mathcal{Z}^{(1)}, \ldots, \mathcal{Z}^{(T_{\max})}$, each of which will be an MDP $\mathcal{Z}^{(t)} = \langle \mathcal{X}^{(t)}, \mathcal{U}^{(t)}, T^{(t)}, R^{(t)}, \gamma \rangle$. Tasks are drawn *i.i.d.* from a fixed, stationary environment; we formalize this assumption in Section 5.4. The goal of the agent is to find the policy parameters $\left\{\boldsymbol{\theta}^{(1)}, \ldots, \boldsymbol{\theta}^{(T_{\max})}\right\}$ that maximize the performance across all tasks: $\frac{1}{T_{\max}} \sum_{t=1}^{T_{\max}} \mathbb{E} \sum_{i=0}^{\infty} \gamma^i R_i^{(t)}$. We do not assume knowledge of the total number of tasks, the order in which tasks will arrive, or the relations between different tasks.

Upon observing each task, the agent will be allowed to interact with the environment for a limited time, typically insufficient for obtaining optimal performance without exploiting information from prior tasks. During this time, the learner will strive to discover any relevant information from the current task to 1) relate it to previously stored knowledge in order to permit transfer and 2) store any newly discovered knowledge for future reuse. At any time, the agent may be evaluated on any previously seen task, so it must retain knowledge from all early tasks in order to perform well.

# 5   Lifelong policy gradient learning

Our framework for lifelong PG learning uses factored representations. The central idea is assuming that the policy parameters for task $t$ can be factored into $\boldsymbol{\theta}^{(t)} \approx \boldsymbol{L}\boldsymbol{s}^{(t)}$, where $\boldsymbol{L} \in \mathbb{R}^{d \times k}$ is a shared dictionary of policy factors and $\boldsymbol{s}^{(t)} \in \mathbb{R}^k$ are task-specific coefficients that select components for the current task. We further assume that we have access to some base PG algorithm that, given a single task, is capable of finding a parametric policy that performs well on the task, although not necessarily optimally.

Upon encountering a new task $\mathcal{Z}^{(t)}$, LPG-FTW (Algorithm 1) will use the base learner to optimize the task-specific coefficients $\boldsymbol{s}^{(t)}$, without modifying the knowledge base $\boldsymbol{L}$. This corresponds to searching for the optimal policy that can be obtained by combining the factors of $\boldsymbol{L}$. Every $M \gg 1$ steps, the agent will update the knowledge base $\boldsymbol{L}$ with any relevant information collected from $\mathcal{Z}^{(t)}$ up to that point. This allows the agent to search for policies with an improved knowledge base in subsequent steps.

---
**Algorithm 1** LPG-FTW$(d, k, \lambda, \mu, M)$

---
$T \leftarrow 0, \quad \boldsymbol{L} \leftarrow$ initializeL$(d, k)$
**loop**
    $t \leftarrow$ getTask()
    **if** isNewTask$(t)$ **then**
        $\boldsymbol{s}^{(t)} \leftarrow$ initializeSt$(k)$
        $T \leftarrow T + 1$
    **else**
        $\boldsymbol{A} \leftarrow \boldsymbol{A} - 2\left(\boldsymbol{s}^{(t)}\boldsymbol{s}^{(t)\top}\right) \otimes \boldsymbol{H}^{(t)}$
        $\boldsymbol{b} \leftarrow \boldsymbol{b} - \boldsymbol{s}^{(t)} \otimes \left(-\boldsymbol{g}^{(t)} + 2\boldsymbol{H}^{(t)}\boldsymbol{\alpha}^{(t)}\right)$
    **for** $i = 1, \ldots, N$ **do**
        $\mathbb{T} \leftarrow$ getTrajectories$(\boldsymbol{L}\boldsymbol{s}^{(t)})$
        $\boldsymbol{s}^{(t)} \leftarrow$ PGStep$(\mathbb{T}, \boldsymbol{L}, \boldsymbol{s}^{(t)}, \mu)$
        **if** $i \bmod M = 0$ **then**
            $\boldsymbol{\alpha}^{(t)} \leftarrow \boldsymbol{L}\boldsymbol{s}^{(t)}$
            $\boldsymbol{g}^{(t)}, \boldsymbol{H}^{(t)} \leftarrow$ gradAndHess$(\boldsymbol{\alpha}^{(t)})$
            $\boldsymbol{A}_{\mathsf{tmp}} \leftarrow \boldsymbol{A} + 2\left(\boldsymbol{s}^{(t)}\boldsymbol{s}^{(t)\top}\right) \otimes \boldsymbol{H}^{(t)}$
            $\boldsymbol{b}_{\mathsf{tmp}} \leftarrow \boldsymbol{b} + \boldsymbol{s}^{(t)} \otimes \left(-\boldsymbol{g}^{(t)} + 2\boldsymbol{H}^{(t)}\boldsymbol{\alpha}^{(t)}\right)$
            $\text{vec}(\boldsymbol{L}) \leftarrow \left(\frac{1}{T}\boldsymbol{A}_{\mathsf{tmp}} - 2\lambda\boldsymbol{I}\right)^{-1}\left(\frac{1}{T}\boldsymbol{b}_{\mathsf{tmp}}\right)$
    $\boldsymbol{A} \leftarrow \boldsymbol{A}_{\mathsf{tmp}}, \quad \boldsymbol{b} \leftarrow \boldsymbol{b}_{\mathsf{tmp}}$

---

Concretely, the agent will strive to solve the following optimization during the training phase:

$$\boldsymbol{s}^{(t)} = \arg\max_{\boldsymbol{s}} \ell(\boldsymbol{L}_{t-1}, \boldsymbol{s}) = \arg\max_{\boldsymbol{s}} \mathcal{J}^{(t)}(\boldsymbol{L}_{t-1}\boldsymbol{s}) - \mu\|\boldsymbol{s}\|_1 \ , \tag{1}$$

where $\boldsymbol{L}_{t-1}$ denotes the $\boldsymbol{L}$ trained up to task $\mathcal{Z}^{(t-1)}$, $\mathcal{J}^{(t)}(\cdot)$ is any PG objective, and the $\ell_1$ norm encourages sparsity. Following Bou Ammar et al. [6], the agent will then solve the following second-

order approximation to the multi-task objective to add knowledge from task $\mathcal{Z}^{(t)}$ into the dictionary:

$$\boldsymbol{L}_t = \arg\max_{\boldsymbol{L}} \hat{g}_t(\boldsymbol{L}) = \arg\max_{\boldsymbol{L}} -\lambda\|\boldsymbol{L}\|_{\mathsf{F}}^2 + \frac{1}{t}\sum_{\hat{t}=1}^{t} \hat{\ell}(\boldsymbol{L}, \boldsymbol{s}^{(\hat{t})}, \boldsymbol{\alpha}^{(\hat{t})}, \boldsymbol{H}^{(\hat{t})}, \boldsymbol{g}^{(\hat{t})}) \;, \qquad (2)$$

where $\hat{\ell}(\boldsymbol{L}, \boldsymbol{s}^{(\hat{t})}, \boldsymbol{\alpha}^{(\hat{t})}, \boldsymbol{H}^{(\hat{t})}, \boldsymbol{g}^{(\hat{t})}) = -\mu\|\boldsymbol{s}^{(\hat{t})}\|_1 + \|\boldsymbol{\alpha}^{(\hat{t})} - \boldsymbol{L}\boldsymbol{s}^{(\hat{t})}\|_{\boldsymbol{H}^{(\hat{t})}}^2 + \boldsymbol{g}^{(\hat{t})\top}(\boldsymbol{L}\boldsymbol{s}^{(\hat{t})} - \boldsymbol{\alpha}^{(\hat{t})})$ is the second-order approximation to the objective of a previously seen task, $\mathcal{Z}^{(\hat{t})}$. The gradient $\boldsymbol{g}^{(\hat{t})} = \nabla_{\boldsymbol{\theta}}\mathcal{J}^{(\hat{t})}(\boldsymbol{\theta})\big|_{\boldsymbol{\theta}=\boldsymbol{\alpha}^{(\hat{t})}}$ and Hessian $\boldsymbol{H}^{(\hat{t})} = \frac{1}{2}\nabla_{\boldsymbol{\theta},\boldsymbol{\theta}^\top}\mathcal{J}^{(\hat{t})}(\boldsymbol{\theta})\big|_{\boldsymbol{\theta}=\boldsymbol{\alpha}^{(\hat{t})}}$ are evaluated at the policy for task $\mathcal{Z}^{(\hat{t})}$ immediately after training, $\boldsymbol{\alpha}^{(\hat{t})} = \boldsymbol{L}_{\hat{t}-1}\boldsymbol{s}^{(\hat{t})}$. The solution to this optimization can be obtained in closed form as $\mathrm{vec}(\boldsymbol{L}_t) = \boldsymbol{A}^{-1}\boldsymbol{b}$, where $\boldsymbol{A} = -2\lambda\boldsymbol{I} + \frac{2}{t}\sum_{\hat{t}=1}^{t}(\boldsymbol{s}^{(\hat{t})}\boldsymbol{s}^{(\hat{t})\top}) \otimes \boldsymbol{H}^{(\hat{t})}$ and $\boldsymbol{b} = \frac{1}{t}\sum_{\hat{t}=1}^{t}\boldsymbol{s}^{(\hat{t})} \otimes (-\boldsymbol{g}^{(\hat{t})} + 2\boldsymbol{H}^{(\hat{t})}\boldsymbol{\alpha}^{(\hat{t})})$. Notably, these can be computed incrementally as each new task arrives, so that $\boldsymbol{L}$ can be updated without preserving data or parameters from earlier tasks. Moreover, the Hessians $\boldsymbol{H}^{(\hat{t})}$ needed to compute $\boldsymbol{A}$ and $\boldsymbol{b}$ can be discarded after each task if the agent is not expected to revisit tasks for further training. If instead the agent will revisit tasks multiple times (e.g., for interleaved multi-task learning), then each $\boldsymbol{H}^{(\hat{t})}$ must be stored at a cost of $O(d^2T_{\mathsf{max}})$.

Intuitively, in Equation 1 the agent leverages knowledge from all past tasks while training on task $\mathcal{Z}^{(t)}$, by searching for $\boldsymbol{\theta}^{(t)}$ in the span of $\boldsymbol{L}_{t-1}$. This makes LPG-FTW fundamentally different from prior multi-model methods that learn each task's parameter vector in isolation and subsequently combine prior knowledge to improve performance. One potential drawback is that, by restricting the search to the span of $\boldsymbol{L}_{t-1}$, we might miss other, potentially better, policies. However, any set of parameters far from the space spanned by $\boldsymbol{L}_{t-1}$ would be uninformative for the multi-task objective, since the approximations to the previous tasks would be poor near the current task's parameters and vice versa. In Equation 2, LPG-FTW approximates the loss around the current set of parameters $\boldsymbol{\alpha}^{(t)}$ via a second-order expansion and finds the $\boldsymbol{L}_t$ that optimizes the average approximate cost over all previously seen tasks, ensuring that the agent does not forget the knowledge required to solve them.

**Time complexity** LPG-FTW introduces an overhead of $O(k \times d)$ per PG step, due to the multiplication of the gradient by $\boldsymbol{L}^\top$. Additionally, every $M \gg 1$ steps, the update step of $\boldsymbol{L}$ takes an additional $O(d^3k^2)$. If the number of parameters $d$ is too high, we could use faster techniques for solving the inverse of $\boldsymbol{A}$ in Equation 2, like the conjugate gradient method, or approximate the Hessian with a Kronecker-factored (KFAC) or diagonal matrix. While we didn't use these approximations, they work well in related methods [6, 29], so we expect LPG-FTW to behave similarly. However, note that the time complexity of LPG-FTW is constant w.r.t. the number of tasks seen, since Equation 2 is solved incrementally. This applies to diagonal matrices, but not to KFAC matrices, which require storing all Hessians and recomputing the cost for every new task, which is infeasible for large numbers of tasks.

## 5.1 Knowledge base initialization

The intuition we have built holds only when a reasonably good $\boldsymbol{L}$ matrix has already been learned. But what happens at the beginning of the learning process, when the agent has not yet seen a substantial number of tasks? If we take the naïve approach of initializing $\boldsymbol{L}$ at random, then the $\boldsymbol{s}^{(t)}$'s are unlikely to be able to find a well-performing policy, and so updates to $\boldsymbol{L}$ will not leverage any useful information.

One common alternative is to initialize the $k$ columns of $\boldsymbol{L}$ with the STL solutions to the first $k$ tasks. However, this method prevents tasks $\mathcal{Z}^{(2)}, \ldots, \mathcal{Z}^{(k)}$ from leveraging information from earlier tasks, impeding them from

---

**Algorithm 2** InitializeL($d, k, \lambda, \mu$)

$T \leftarrow 0, \quad \boldsymbol{L} \leftarrow$ empty($d, 0$)
**while** $T < k$ **do**
    $t \leftarrow$ getTask()
    $\boldsymbol{s}^{(t)} \leftarrow$ initializeSt($k$)
    $T \leftarrow T + 1$
    **for** $i = 1, \ldots, N$ **do**
        $\mathbb{T} \leftarrow$ getTrajectories($\boldsymbol{L}\boldsymbol{s}^{(t)}$)
        $\boldsymbol{s}^{(t)}, \boldsymbol{\epsilon}^{(t)} \leftarrow$ PGStep($\mathbb{T}, \boldsymbol{L}, \boldsymbol{s}^{(t)}, \boldsymbol{\epsilon}^{(t)}, \mu$)
    $\boldsymbol{L} \leftarrow$ addColumn($\boldsymbol{L}, \boldsymbol{\epsilon}^{(t)}$)
    $\boldsymbol{\alpha}^{(t)} \leftarrow \boldsymbol{L}\boldsymbol{s}^{(t)} + \boldsymbol{\epsilon}^{(t)}$
    $\boldsymbol{g}^{(t)}, \boldsymbol{H}^{(t)} \leftarrow$ gradAndHess($\boldsymbol{\alpha}^{(t)}$)
    $\boldsymbol{A} \leftarrow \boldsymbol{A} + 2\left(\boldsymbol{s}^{(t)}\boldsymbol{s}^{(t)\top}\right) \otimes \boldsymbol{H}^{(t)}$
    $\boldsymbol{b} \leftarrow \boldsymbol{b} + \boldsymbol{s}^{(t)} \otimes \left(-\boldsymbol{g}^{(t)} + 2\boldsymbol{H}^{(t)}\boldsymbol{\alpha}^{(t)}\right)$

---

achieving potentially higher performance. Moreover, several tasks might rediscover information, leading to wasted training time and capacity of $\boldsymbol{L}$.

We propose an initialization method (Algorithm 2) that enables early tasks to leverage knowledge from previous tasks and prevents the discovery of redundant information. The algorithm starts from

an empty dictionary and adds error vectors $\boldsymbol{\epsilon}^{(t)} \in \mathbb{R}^d$ for the initial $k$ tasks. For each task $\mathcal{Z}^{(t)}$, we modify Equation 1 for learning $\boldsymbol{s}^{(t)}$ by adding $\boldsymbol{\epsilon}^{(t)}$ as additional learnable parameters:

$$\boldsymbol{s}^{(t)}, \boldsymbol{\epsilon}^{(t)} = \arg\max_{\boldsymbol{s}, \boldsymbol{\epsilon}} \mathcal{J}^{(t)}(\boldsymbol{L}_{t-1}\boldsymbol{s} + \boldsymbol{\epsilon}) - \mu\|\boldsymbol{s}\|_1 - \lambda\|\boldsymbol{\epsilon}\|_2^2 \ .$$

Each $\boldsymbol{\epsilon}^{(t)}$ finds knowledge of task $\mathcal{Z}^{(t)}$ not contained in $\boldsymbol{L}$ and is then incorporated as a new column of $\boldsymbol{L}$. Note that this initialization process replaces the standard PG training of tasks $\mathcal{Z}^{(1)}, \dots, \mathcal{Z}^{(k)}$, and therefore does not require collecting additional data beyond that required by the base PG method.

## 5.2 Base policy gradient algorithms

Now, we show how two STL PG learning algorithms can be used as the base learner of LPG-FTW.

### 5.2.1 Episodic REINFORCE

Episodic REINFORCE updates parameters as $\boldsymbol{\theta}_j \leftarrow \boldsymbol{\theta}_{j-1} + \eta_j \boldsymbol{g}_{\boldsymbol{\theta}_{j-1}}$, with the policy gradient given by $\boldsymbol{g}_{\boldsymbol{\theta}} = \nabla_{\boldsymbol{\theta}} \mathcal{J}(\boldsymbol{\theta}) = \mathbb{E}\left[\sum_{i=0}^{\infty} \nabla_{\boldsymbol{\theta}} \log \pi_{\boldsymbol{\theta}}(\boldsymbol{x}_i, \boldsymbol{u}_i) A(\boldsymbol{x}_i, \boldsymbol{u}_i)\right]$, where $A(\boldsymbol{x}, \boldsymbol{u})$ is the advantage function. LPG-FTW would then update the $\boldsymbol{s}^{(t)}$'s as $\boldsymbol{s}_j^{(t)} \leftarrow \boldsymbol{s}_{j-1}^{(t)} + \eta_j \nabla_{\boldsymbol{s}}\left[\mathcal{J}^{(t)}(\boldsymbol{L}_{t-1}\boldsymbol{s}) - \mu\|\boldsymbol{s}\|_1\right]\big|_{\boldsymbol{s}=\boldsymbol{s}_j^{(t)}}$, with the gradient given by $\nabla_{\boldsymbol{s}}\left[\mathcal{J}^{(t)}(\boldsymbol{L}_{t-1}\boldsymbol{s}) - \mu\|\boldsymbol{s}\|_1\right] = \boldsymbol{L}_{t-1}^{\top}\boldsymbol{g}_{\boldsymbol{L}_{t-1}\boldsymbol{s}} - \mu\,\mathrm{sign}(\boldsymbol{s})$. The Hessian for Equation 2 is given by $\boldsymbol{H} = \frac{1}{2}\mathbb{E}\left[\sum_{i=0}^{\infty} \nabla_{\boldsymbol{\theta},\boldsymbol{\theta}^{\top}} \log \pi_{\boldsymbol{\theta}}(\boldsymbol{x}_i, \boldsymbol{u}_i) A(\boldsymbol{x}_i, \boldsymbol{u}_i)\right]$, which evaluates to $\boldsymbol{H} = -\frac{1}{2\sigma^2}\mathbb{E}\left[\sum_{i=0}^{\infty} \boldsymbol{x}\boldsymbol{x}^{\top} A(\boldsymbol{x}_i, \boldsymbol{u}_i)\right]$ in the case where the policy is a linear Gaussian (i.e., $\pi_{\boldsymbol{\theta}} = \mathcal{N}(\boldsymbol{\theta}^{\top}\boldsymbol{x}, \sigma)$). One major drawback of this is that the Hessian is not negative definite, so Equation 2 might move the policy arbitrarily far from the original policy used for sampling trajectories.

### 5.2.2 Natural Policy Gradient

The natural PG (NPG) algorithm allows us to get around this issue. We use the formulation followed by Rajeswaran et al. [28], which at each iteration optimizes $\max_{\boldsymbol{\theta}} \boldsymbol{g}_{\boldsymbol{\theta}_{j-1}}^{\top}(\boldsymbol{\theta} - \boldsymbol{\theta}_{j-1})$ subject to the quadratic constraint $\|\boldsymbol{\theta} - \boldsymbol{\theta}_{j-1}\|_{\boldsymbol{F}_{\boldsymbol{\theta}_{j-1}}}^2 \leq \delta$, where $\boldsymbol{F}_{\boldsymbol{\theta}} = \mathbb{E}\left[\nabla_{\boldsymbol{\theta}} \log \pi_{\boldsymbol{\theta}}(\boldsymbol{x}, \boldsymbol{u}) \nabla_{\boldsymbol{\theta}} \log \pi_{\boldsymbol{\theta}}(\boldsymbol{x}, \boldsymbol{u})^{\top}\right]$ is the approximate Fisher information of $\pi_{\boldsymbol{\theta}}$ [20]. The base learner would then update the policy parameters at each iteration as $\boldsymbol{\theta}_j \leftarrow \boldsymbol{\theta}_{j-1} + \eta_{\boldsymbol{\theta}} \boldsymbol{F}_{\boldsymbol{\theta}_{j-1}}^{-1} \boldsymbol{g}_{\boldsymbol{\theta}_{j-1}}$, with $\eta_{\boldsymbol{\theta}} = \sqrt{\delta/(\boldsymbol{g}_{\boldsymbol{\theta}_{j-1}}^{\top} \boldsymbol{F}_{\boldsymbol{\theta}_{j-1}}^{-1} \boldsymbol{g}_{\boldsymbol{\theta}_{j-1}})}$.

To incorporate NPG as the base learner in LPG-FTW, at each step we solve $\max_{\boldsymbol{s}} \boldsymbol{g}_{\boldsymbol{s}_{j-1}^{(t)}}^{\top}(\boldsymbol{s} - \boldsymbol{s}_{j-1}^{(t)})$ subject to $\|\boldsymbol{s} - \boldsymbol{s}_{j-1}^{(t)}\|_{\boldsymbol{F}_{\boldsymbol{s}_{j-1}^{(t)}}}^2 \leq \delta$, which gives us the update: $\boldsymbol{s}_j^{(t)} \leftarrow \boldsymbol{s}_{j-1}^{(t)} + \eta_{\boldsymbol{s}^{(t)}} \boldsymbol{F}_{\boldsymbol{s}_{j-1}^{(t)}}^{-1} \boldsymbol{g}_{\boldsymbol{s}_{j-1}^{(t)}}$. We compute the Hessian for Equation 2 as $\boldsymbol{H} = -\frac{1}{\eta_{\boldsymbol{\theta}}} \boldsymbol{F}_{\boldsymbol{\theta}_{j-1}}$ using the equivalent soft-constrained problem: $\widehat{\mathcal{J}}(\boldsymbol{\theta}) = \boldsymbol{g}_{\boldsymbol{\theta}_{j-1}}^{\top}(\boldsymbol{\theta} - \boldsymbol{\theta}_{j-1}) + \frac{\|\boldsymbol{\theta} - \boldsymbol{\theta}_{j-1}\|_{\boldsymbol{F}_{\boldsymbol{\theta}_{j-1}}}^2 - \delta}{2\eta_{\boldsymbol{\theta}}}$. This Hessian *is* negative definite, and thus encourages the parameters to stay close to the original ones, where the approximation is valid.

## 5.3 Connections to PG-ELLA

LPG-FTW and PG-ELLA [6] both learn a factorization of policies into $\boldsymbol{L}$ and $\boldsymbol{s}^{(t)}$. To optimize the factors, PG-ELLA first trains individual task policies via STL, potentially leading to policy parameters that are incompatible with a shared $\boldsymbol{L}$. In contrast, our method learns the $\boldsymbol{s}^{(t)}$'s directly via PG learning, leveraging shared knowledge in $\boldsymbol{L}$ to accelerate the learning and restricting the $\boldsymbol{\alpha}^{(t)}$'s to the span of $\boldsymbol{L}$. This choice implies that, even if we find the (often infeasible) optimal $\boldsymbol{s}^{(t)}$, this may not result in an optimal policy, so we explicitly add a linear term in Equation 2 omitted in PG-ELLA. We also improve PG-ELLA's knowledge base initialization, which typically uses the policies of the first $k$ tasks. Instead, LPG-FTW exploits the policies from the few previously observed tasks to 1) accelerate the learning of the earliest tasks and 2) discover $k$ distinct knowledge factors. These improvements enable our method to operate in a true lifelong setting, where the agent encounters tasks sequentially. In contrast, PG-ELLA was evaluated in the easier interleaved multi-task setting, in which the agent experiences each task multiple times, alternating between tasks frequently. These modifications also enable applying LPG-FTW to far more complex dynamical systems than PG-ELLA, including domains requiring deep policies, previously out of reach for factored policy learning methods.

## 5.4 Theoretical guarantees

We now show that LPG-FTW converges to the optimal multi-task objective for any ordering over tasks, despite the online approximation of keeping the $s^{(t)}$'s fixed after initial training. We substantially adapt the proofs by Ruvolo and Eaton [32] to handle the non-optimality of the $\alpha^{(t)}$'s and the fact that the $s^{(t)}$'s and $L$ optimize different objectives. Complete proofs are available in Appendix A.

The objective defined in Equation 2, $\hat{g}$, considers the optimization of each $s^{(t)}$ separately with the $L_t$ known up to that point, and is a surrogate for our actual objective:

$$g_t(\boldsymbol{L}) = \frac{1}{t}\sum_{\hat{t}=1}^{t}\max_{\boldsymbol{s}^{(\hat{t})}}\left\{\|\boldsymbol{\alpha}^{(\hat{t})} - \boldsymbol{L}\boldsymbol{s}^{(\hat{t})}\|^2_{\boldsymbol{H}^{(\hat{t})}} + {\boldsymbol{g}^{(\hat{t})}}^\top(\boldsymbol{L}\boldsymbol{s}^{(\hat{t})} - \boldsymbol{\alpha}^{(\hat{t})}) - \mu\|\boldsymbol{s}^{(\hat{t})}\|_1\right\} - \lambda\|\boldsymbol{L}\|^2_{\mathsf{F}} \;,$$

which considers the simultaneous optimization of all $s^{(t)}$'s. We define the expected objective as:

$$g(\boldsymbol{L}) = \mathbb{E}_{\boldsymbol{H}^{(t)},\boldsymbol{g}^{(t)},\boldsymbol{\alpha}^{(t)}}\left[\max_{\boldsymbol{s}}\hat{\ell}(\boldsymbol{L},\boldsymbol{s},\boldsymbol{\alpha}^{(t)},\boldsymbol{H}^{(t)},\boldsymbol{g}^{(t)})\right] \;,$$

which represents how well a particular $\boldsymbol{L}$ can represent a randomly selected task without modifications.

We show that 1) $\boldsymbol{L}_t$ becomes increasingly stable, 2) $\hat{g}_t$, $g_t$, and $g$ converge to the same value, and 3) $\boldsymbol{L}_t$ converges to a stationary point of $g$. These results are based on the following assumptions:

A. The tuples $(\boldsymbol{H}^{(t)},\boldsymbol{g}^{(t)})$ are drawn *i.i.d.* from a distribution with compact support.
B. The sequence $\{\boldsymbol{\alpha}^{(t)}\}_{t=1}^{\infty}$ is stationary and $\phi$-mixing.
C. The magnitude of $\mathcal{J}^{(t)}(\boldsymbol{0})$ is bounded by $B$.
D. For all $\boldsymbol{L}$, $\boldsymbol{H}^{(t)}$, $\boldsymbol{g}^{(t)}$, and $\boldsymbol{\alpha}^{(t)}$, the largest eigenvalue (smallest in magnitude) of $\boldsymbol{L}_\gamma^\top\boldsymbol{H}^{(t)}\boldsymbol{L}_\gamma$ is at most $-\kappa$, with $\kappa > 0$, where $\gamma$ is the set of non-zero indices of $\boldsymbol{s}^{(t)} = \arg\max_{\boldsymbol{s}}\hat{\ell}(\boldsymbol{L},\boldsymbol{s},\boldsymbol{H}^{(t)},\boldsymbol{g}^{(t)},\boldsymbol{\alpha}^{(t)})$. The non-zero elements of the unique maximizing $\boldsymbol{s}^{(t)}$ are given by: $\boldsymbol{s}_\gamma^{(t)} = (\boldsymbol{L}_\gamma^\top\boldsymbol{H}^{(t)}\boldsymbol{L}_\gamma)^{-1}\left(\boldsymbol{L}^\top\left(\boldsymbol{H}^{(t)}\boldsymbol{\alpha}^{(t)} - \boldsymbol{g}^{(t)}\right) - \mu\,\mathrm{sign}\left(\boldsymbol{s}_\gamma^{(t)}\right)\right)$.

**Proposition 1.** $\boldsymbol{L}_t - \boldsymbol{L}_{t-1} = O(\frac{1}{t})$ .

*Proof sketch.* First, we show that the entries of $\boldsymbol{L}$, $\boldsymbol{s}^{(t)}$, and $\boldsymbol{\alpha}^{(t)}$ are bounded by Assumptions A and C and the regularization terms. Next, we show that $\hat{g}_t - \hat{g}_{t-1}$ is $O\left(\frac{1}{t}\right)$–Lipschitz. We finish the proof with the facts that $\boldsymbol{L}_{t-1}$ maximizes $\hat{g}_{t-1}$ and the eigenvalues of the Hessian of $\hat{g}_{t-1}$ are bounded. ∎

The critical step for adapting the proof from Ruvolo and Eaton [32] to our algorithm is to introduce the following lemma, which shows the equality of the maximizers of $\ell$ and $\hat{\ell}$.

**Lemma 1.** $\hat{\ell}(\boldsymbol{L}_t,\boldsymbol{s}^{(t+1)},\boldsymbol{\alpha}^{(t+1)},\boldsymbol{H}^{(t+1)},\boldsymbol{g}^{(t+1)}) = \max_{\boldsymbol{s}}\hat{\ell}(\boldsymbol{L}_t,\boldsymbol{s},\boldsymbol{\alpha}^{(t+1)},\boldsymbol{H}^{(t+1)},\boldsymbol{g}^{(t+1)})$ .

*Proof sketch.* The fact that the single-task objective $\hat{\ell}$ is a second-order approximation at $\boldsymbol{s}^{(t+1)}$ of $\ell$, along with the fact that $\boldsymbol{s}^{(t+1)}$ is a maximizer of $\ell$, implies that $\boldsymbol{s}^{(t+1)}$ is also a maximizer of $\hat{\ell}$. ∎

**Proposition 2.**    *1. $\hat{g}_t(\boldsymbol{L}_t)$ converges a.s.*     *3. $g_t(\boldsymbol{L}_t) - \hat{g}(\boldsymbol{L}_t)$ converges a.s. to 0*
                     *2. $g_t(\boldsymbol{L}_t) - \hat{g}_t(\boldsymbol{L}_t)$ converges a.s. to 0*   *4. $g(\boldsymbol{L}_t)$ converges a.s.*

*Proof sketch.* First, using Lemma 1, we show that the sum of negative variations of the stochastic process $u_t = \hat{g}_t(\boldsymbol{L}_t)$ is bounded. Given this result, we show that $u_t$ is a quasi-martingale that converges almost surely (Part 1). This fact, along with a simple lemma of positive sequences, allows us to prove Part 2. The final two parts can be shown due to the equivalence of $g$ and $g_t$ as $t \to \infty$. ∎

**Proposition 3.** *The distance between $\boldsymbol{L}_t$ and the set of all stationary points of $g$ converges a.s. to 0.*
*Proof sketch.* We use the fact that $\hat{g}_t$ and $g$ have Lipschitz gradients with constants independent of $t$. This fact, combined with the fact that $\hat{g}_t$ and $g$ converge almost surely completes the proof. ∎

## 6 Experimental evaluation

We evaluated our method on a range of complex continuous control domains, showing a substantial increase in learning speed and a dramatic reduction in catastrophic forgetting. See Appendix B for additional details. Code and training videos are available at `github.com/GRASP-ML/LPG-FTW` and on the last author's website.

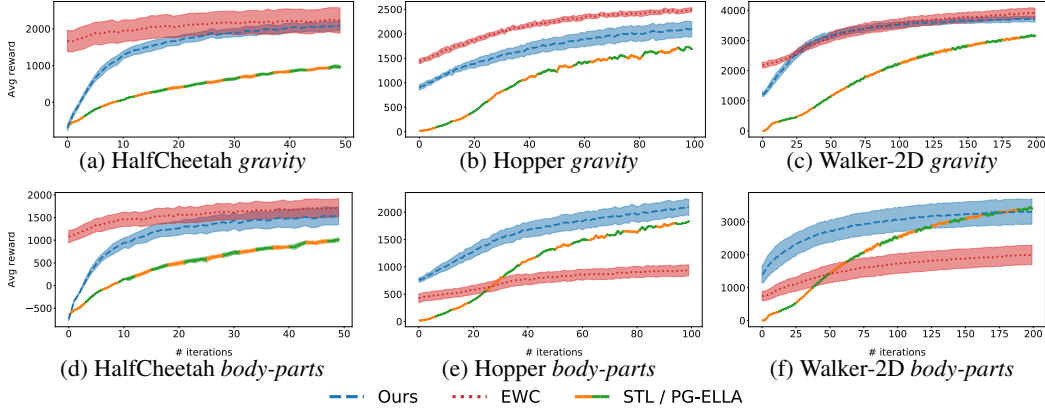

Figure 1: Average performance during training across all tasks for six MuJoCo domains. LPG-FTW is consistently faster than STL and PG-ELLA (which by definition learn at the same pace) in achieving proficiency, and achieves better final performance in five domains and equivalent performance in the remaining one. EWC is faster and converges to higher performance than LPG-FTW in some domains, but completely fails to learn in others. Shaded error bars denote standard error over five random task orderings and parameter initializations.

**Baselines**   We compared against STL, which does not transfer knowledge across tasks, using NPG as described in Section 5.2.2. We then chose EWC [21] from the single-model family, which places a quadratic penalty for deviating from earlier tasks' parameters. Finally, we compared against PG-ELLA, described in Section 5.3. All lifelong algorithms used NPG as the base learning method.

**Evaluation procedure**   We chose the hyper-parameters of NPG to maximize the performance of STL on a single task, and used those hyper-parameters for all agents. For EWC, we searched for the regularization parameter over five tasks on each domain. For LPG-FTW and PG-ELLA, we fixed all regularization parameters to $10^{-5}$ and the number of columns in $\boldsymbol{L}$ to $k=5$, unless otherwise noted. In LPG-FTW, we used the simplest setting for the update schedule of $\boldsymbol{L}$, $M=N$. All experiments were repeated over five trials with different random seeds for parameter initialization and task ordering.

### 6.1   Empirical evaluation on OpenAI Gym MuJoCo domains

We first evaluated LPG-FTW on simple MuJoCo environments from OpenAI Gym [38, 7]. We selected the HalfCheetah, Hopper, and Walker-2D environments, and created two different evaluation domains for each: a *gravity* domain, where each task corresponded to a random gravity value between $0.5g$ and $1.5g$, and a *body-parts* domain, where the size and mass of each of four parts of the body (head, torso, thigh, and leg) was randomly set to a value between $0.5\times$ and $1.5\times$ its nominal value. These choices led to highly diverse tasks, as we show in Appendix C. We generated tasks using the `gym-extensions` [16] package, but modified it so each body part was scaled independently.

We created $T_{\max} = 20$ tasks for HalfCheetah and Hopper domains, and $T_{\max} = 50$ for Walker-2D domains. The agents were allowed to train on each task for a fixed number of iterations before moving on to the next. For these simple experiments, all agents used linear policies. For the Walker-2D *body-parts* domain, we set the capacity of $\boldsymbol{L}$ to $k = 10$, since we found empirically that it required a higher capacity. The NPG hyper-parameters were tuned without body-parts or gravity modifications.

Figure 1 shows the average performance over all tasks as a function of the NPG training iterations. LPG-FTW consistently learned faster than STL, and obtained higher final performance on five out of the six domains. Learning the task-specific coefficients $\boldsymbol{s}^{(t)}$ directly via policy search increased the learning speed of LPG-FTW, whereas PG-ELLA was limited to the learning speed of STL, as indicated by the shared learning curves. EWC was faster than LPG-FTW in reaching high-performing policies in four domains, primarily due to the fact that EWC uses a single shared policy across all tasks, which enables it to have starting policies with high performance. However, EWC failed to even match the STL performance in two of the domains. We hypothesize that this was due to the fact that the tasks were highly varied (particularly in the *body-parts* domains, since there were four

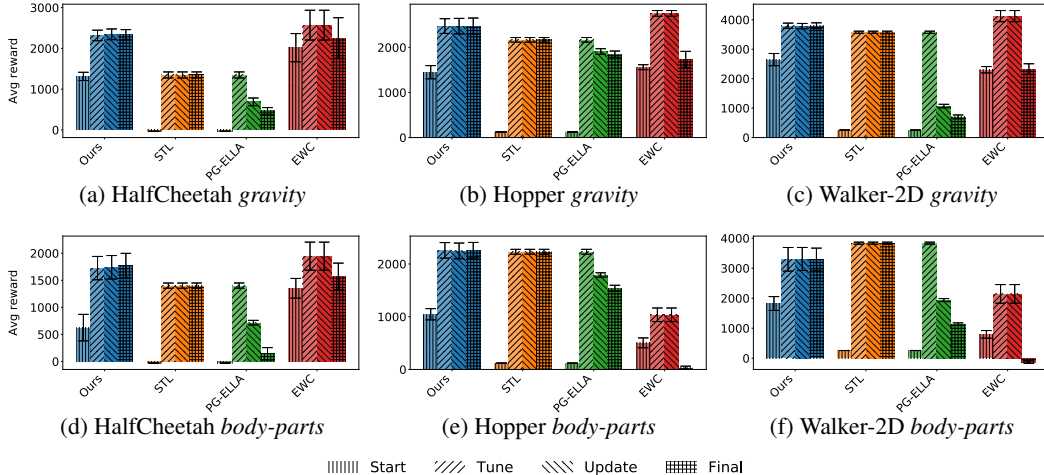

| (a) HalfCheetah *gravity* | (b) Hopper *gravity* | (c) Walker-2D *gravity* |
| (d) HalfCheetah *body-parts* | (e) Hopper *body-parts* | (f) Walker-2D *body-parts* |

|||||||| Start    ////  Tune    \\\\ Update    ##### Final

Figure 2: Average performance at the beginning of training (start), after all training iterations (tune, equivalent to the final point in Figure 1), after the update step for PG-ELLA and LPG-FTW (update), and after all tasks have been trained (final). The update step in LPG-FTW never hinders performance, and even after all tasks have been trained performance is maintained. PG-ELLA always performed worse than STL. EWC suffered from catastrophic forgetting in five domains, in two resulting in degradation below initial performance. Error bars denote standard error over five random task orderings and parameter initializations.

different axes of variation), and the single shared policy was unable to perform well in all domains. Appendix D shows an evaluation with various versions of EWC attempting to alleviate these issues.

Results in Figure 1 consider only how fast the agent learns a new task using information from earlier tasks. PG-ELLA and LPG-FTW then perform an update step (Equation 2 for LPG-FTW) where they incorporate knowledge from the current task into $L$. The third bar from the left per each algorithm in Figure 2 shows the average performance after this step, revealing that LPG-FTW maintained performance, whereas PG-ELLA's performance decreased. This is because LPG-FTW ensures that the approximate objective is computed near points in the parameter space that the current basis $L$ can generate, by finding $\boldsymbol{\alpha}^{(t)}$ via a search over the span of $L$. A critical component of lifelong learning algorithms is their ability to avoid catastrophic forgetting. To evaluate the capacity of LPG-FTW to retain knowledge from earlier tasks, we evaluated the policies obtained from the knowledge base $L$ trained on all tasks, without modifying the $\boldsymbol{s}^{(t)}$'s. The rightmost bar in each algorithm in Figure 2 shows the average final performance across all tasks. LPG-FTW successfully retained knowledge of all tasks, showing no signs of catastrophic forgetting on any of the domains. The PG-ELLA baseline suffered from forgetting in all domains, and EWC in all but one of the domains. Moreover, the final performance of LPG-FTW was the best among all baselines in all but one domain.

## 6.2 Empirical evaluation on more challenging Meta-World domains

Results so far show that our method improves performance and completely avoids forgetting in simple settings. To showcase the flexibility of our framework, we evaluated it on Meta-World [41], a substantially more challenging benchmark, whose tasks involve using a simulated Sawyer robotic arm to manipulate various objects in diverse ways, and have been shown to be notoriously difficult for state-of-the-art multi-task learning and meta-learning algorithms. Concretely, we evaluated all methods on sequential versions of the MT10 benchmark, with $T_{\max} = 10$ tasks, and the MT50 benchmark, using a subset of $T_{\max} = 48$ tasks. We added an experience replay (ER) baseline that uses importance sampling over a replay buffer from all previous tasks' data to encourage knowledge retention, with a 50-50 replay rate as suggested by Rolnick et al. [30]. We chose the NPG hyper-parameters on the `reach` task, which is the simplest task from the benchmark. For LPG-FTW and PG-ELLA, we fixed the number of latent components as $k = 3$. All algorithms used a Gaussian policy parameterized by a multi-layer perceptron with two hidden layers of 32 units and `tanh` activation. We also evaluated EWC with a higher capacity (EWC_h), with 50 hidden units for MT10 and 40 for MT50, to ensure that it was given access to approximately the same number of parameters as our

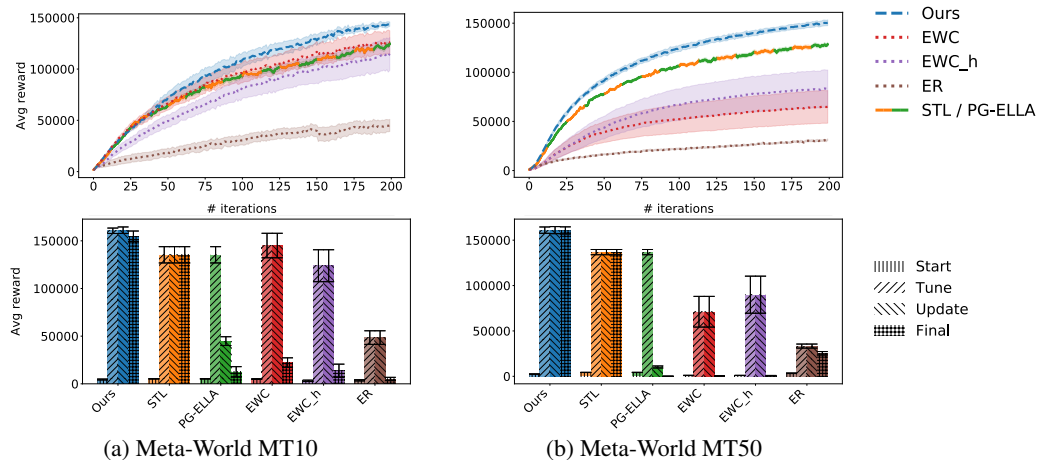

(a) Meta-World MT10          (b) Meta-World MT50

Figure 3: Performance on the Meta-World benchmark. Top: average performance during training across all tasks. Bottom: average performance at the beginning of training (start), after all training iterations (tune), after the update step for PG-ELLA and LPG-FTW (update), and after all tasks have been trained (final). In this notoriously challenging benchmark, LPG-FTW still improves the performance of STL and all baselines, and suffers from no catastrophic forgetting.

method. Given the high diversity of the tasks considered in this evaluation, we allowed all algorithms to use task-specific output layers, in order to specialize policies to each individual task.

The top row of Figure 3 shows average learning curves across tasks. LPG-FTW again was faster in training, showing that the restriction that the agent only train the $s^{(t)}$'s for each new task does not harm its ability to solve complex, highly diverse problems. The difference in learning speed was particularly noticeable in MT50, where single-model methods became saturated. To our knowledge, this is the first time lifelong transfer has been shown on the challenging Meta-World benchmark. The bottom row of Figure 3 shows that LPG-FTW suffered from a small amount of forgetting on MT10. However, on MT50, where $L$ trained on sufficient tasks for convergence, our method suffered from no forgetting. In contrast, none of the baselines was capable of accelerating the learning, and they all suffered from dramatic forgetting, particularly on MT50, when needing to learn more tasks. Adding capacity to EWC did not substantially alter these results, showing that our method's ability to handle highly varied tasks does not stem from its higher capacity from using $k$ factors, but instead to the use of *different* models for each task by intelligently combining the shared latent policy factors.

## Conclusion

We proposed a method for lifelong PG learning that enables RL learners to quickly learn to solve new tasks by leveraging knowledge accumulated from earlier tasks. We showed empirically that our method, LPG-FTW, does not suffer from catastrophic forgetting, and therefore permits learning a large number of tasks in sequence. Moreover, we prove theoretically that our algorithm is guaranteed to converge to the approximate multi-task objective, despite operating completely online.

Our method can be viewed as an improvement over the popular PG-ELLA algorithm [6], which to date had not been applicable to highly complex and diverse RL problems like we study in this work, especially in our evaluation on the Meta-World benchmark.

Two of the primary limitations of our work are the reliance on task indicators $(t)$ to reconstruct individual task policies via the $s^{(t)}$'s, and the assumption that the world is stationary and tasks are drawn *i.i.d.* One possible way to address the former challenge is to assume each new batch of experiences comes from a new task, as has been done in prior work [25]. However, note that this would require a small amount of retraining at evaluation time, since the agent would need to discover which task is currently being tested. To address the latter challenge of non-stationarity, our initialization method in Section 5.1 could be extended to dynamically add and remove factors as the environment shifts over time. We leave this avenue of investigation open for future work.

## Broader Impact

One of the key contributions of our work is reducing the amount of experience required by RL agents to achieve proficiency at a multitude of tasks. The method we present here is a first plausible solution to solving a highly diverse set of RL tasks in a lifelong setting. Research in this direction that further reduces the amount of experience required to learn proficient policies would enable RL training on systems where experience is expensive, such as training real robotic systems or learning policies for medical treatments. In these settings, training RL policies has been impractical to date, but could potentially have a large positive impact by discovering policies superior to those conceivable by human experts with domain knowledge.

## Acknowledgments and Disclosure of Funding

We would like to thank Kyle Vedder and the anonymous reviewers for their valuable feedback on this work. The research presented in this paper was partially supported by the DARPA Lifelong Learning Machines program under grant FA8750-18-2-0117, by the DARPA SAIL-ON program under contract HR001120C0040, and by ARO MURI grant W911NF-20-1-0080. The views and conclusions in this paper are those of the authors and should not be interpreted as representing the official policies, either expressed or implied, of the Defense Advanced Research Projects Agency, the Army Research Office, or the U.S. Government. The U.S. Government is authorized to reproduce and distribute reprints for Government purposes notwithstanding any copyright notation herein.

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
