[Supplementary Material]

# Appendices to
# "Lifelong Policy Gradient Learning of Factored Policies for Faster Training Without Forgetting"

by **Jorge A. Mendez**, **Boyu Wang**, and **Eric Eaton**

## A   Proofs of theoretical guarantees

Here, we present complete proofs for the three results on the convergence of LPG-FTW described in Section 5.3 of the main paper. First, recall the definitions of the actual objective we want to maximize:

$$g_t(\boldsymbol{L}) = \frac{1}{t} \sum_{\hat{t}=1}^{t} \max_{\boldsymbol{s}^{(\hat{t})}} \left\{ \|\boldsymbol{\alpha}^{(\hat{t})} - \boldsymbol{L}\boldsymbol{s}^{(\hat{t})}\|_{\boldsymbol{H}^{(\hat{t})}}^2 + {\boldsymbol{g}^{(\hat{t})}}^{\top}(\boldsymbol{L}\boldsymbol{s}^{(\hat{t})} - \boldsymbol{\alpha}^{(\hat{t})}) - \mu\|\boldsymbol{s}^{(\hat{t})}\|_1 \right\} - \lambda\|\boldsymbol{L}\|_{\mathsf{F}}^2 \ ,$$

the surrogate objective we use for optimizing $\boldsymbol{L}$:

$$\hat{g}_t(\boldsymbol{L}) = -\lambda\|\boldsymbol{L}\|_{\mathsf{F}}^2 + \frac{1}{t}\sum_{\hat{t}=1}^{t}\hat{\ell}(\boldsymbol{L}, \boldsymbol{s}^{(\hat{t})}, \boldsymbol{\alpha}^{(\hat{t})}, \boldsymbol{H}^{(\hat{t})}, \boldsymbol{g}^{(\hat{t})}) \ ,$$

and the expected objective:

$$g(\boldsymbol{L}) = \mathbb{E}_{\boldsymbol{H}^{(t)}, \boldsymbol{g}^{(t)}, \boldsymbol{\alpha}^{(t)}} \left[ \max_{\boldsymbol{s}} \hat{\ell}(\boldsymbol{L}, \boldsymbol{s}, \boldsymbol{\alpha}^{(t)}, \boldsymbol{H}^{(t)}, \boldsymbol{g}^{(t)}) \right] \ ,$$

with $\hat{\ell}(\boldsymbol{L}, \boldsymbol{s}, \boldsymbol{\alpha}, \boldsymbol{H}, \boldsymbol{g}) = -\mu\|\boldsymbol{s}\|_1 + \|\boldsymbol{\alpha} - \boldsymbol{L}\boldsymbol{s}\|_{\boldsymbol{H}}^2 + \boldsymbol{g}^{\top}(\boldsymbol{L}\boldsymbol{s} - \boldsymbol{\alpha})$. The convergence results of LPG-FTW are summarized as: 1) the knowledge base $\boldsymbol{L}_t$ becomes increasingly stable, 2) $\hat{g}_t$, $g_t$, and $g$ converge to the same value, and 3) $\boldsymbol{L}_t$ converges to a stationary point of $g$. These results, given below as Propositions 1, 2, and 3, are based on the following assumptions:

    A.  The tuples $\left(\boldsymbol{H}^{(t)}, \boldsymbol{g}^{(t)}\right)$ are drawn *i.i.d.* from a distribution with compact support.

    B.  The sequence $\{\boldsymbol{\alpha}^{(t)}\}_{t=1}^{\infty}$ is stationary and $\phi$-mixing.

    C.  The magnitude of $\mathcal{J}^{(t)}(\boldsymbol{0})$ is bounded by $B$.

    D.  For all $\boldsymbol{L}$, $\boldsymbol{H}^{(t)}$, $\boldsymbol{g}^{(t)}$, and $\boldsymbol{\alpha}^{(t)}$, the largest eigenvalue (smallest in magnitude) of $\boldsymbol{L}_{\gamma}^{\top}\boldsymbol{H}^{(t)}\boldsymbol{L}_{\gamma}$ is at most $-2\kappa$, with $\kappa > 0$, where $\gamma$ is the set of non-zero indices of $\boldsymbol{s}^{(t)} = \arg\max_{\boldsymbol{s}}\hat{\ell}(\boldsymbol{L}, \boldsymbol{s}, \boldsymbol{H}^{(t)}, \boldsymbol{g}^{(t)}, \boldsymbol{\alpha}^{(t)})$. The non-zero elements of the unique maximizing $\boldsymbol{s}^{(t)}$ are given by: $\boldsymbol{s}_{\gamma}^{(t)} = \left(\boldsymbol{L}_{\gamma}^{\top}\boldsymbol{H}^{(t)}\boldsymbol{L}_{\gamma}\right)^{-1}\left(\boldsymbol{L}^{\top}\left(\boldsymbol{H}^{(t)}\boldsymbol{\alpha}^{(t)} - \boldsymbol{g}^{(t)}\right) - \mu\operatorname{sign}\left(\boldsymbol{s}_{\gamma}^{(t)}\right)\right)$.

Note that the $\boldsymbol{\alpha}^{(t)}$'s are not independently obtained, so we cannot assume they are *i.i.d.* like Ruvolo and Eaton [32]. Therefore, we use a weaker assumption on the sequence of $\boldsymbol{\alpha}^{(t)}$'s found by our algorithm, which enables us to use the Donsker theorem [2] and the Glivenko-Cantelli theorem [1].

**Claim 1.** *$\exists\, c_1, c_2, c_3 \in \mathbb{R}$ such that no element of $\boldsymbol{L}_t$, $\boldsymbol{s}^{(t)}$, and $\boldsymbol{\alpha}^{(t)}$ has magnitude greater than $c_1$, $c_2$, and $c_3$, respectively, $\forall t \in \{1, \ldots, \infty\}$.*

*Proof.* We complete this proof by strong induction. In the base case, $\boldsymbol{L}_1$ is given by $\arg\max_{\boldsymbol{\epsilon}} \mathcal{J}^{(1)}(\boldsymbol{\epsilon}) - \lambda\|\boldsymbol{\epsilon}\|_2^2$. If $\boldsymbol{\epsilon} = \boldsymbol{0}$, the objective becomes $\mathcal{J}^{(1)}(\boldsymbol{0})$, which is bounded by Assumption C. This implies that if $\boldsymbol{\epsilon}$ grows too large, $-\lambda\|\boldsymbol{\epsilon}\|_2$ would be too negative, and then it would not be a maximizer. $\boldsymbol{s}^{(1)} = 1$ per Algorithm 2, and so $\boldsymbol{\alpha}^{(1)} = \boldsymbol{L}_1$, which we just showed is bounded.

Then, for $t \leq k$, we have that $\boldsymbol{s}^{(t)}$ and $\boldsymbol{\epsilon}^{(t)}$ are given by $\arg\max_{\boldsymbol{s}, \boldsymbol{\epsilon}} \mathcal{J}^{(t)}(\boldsymbol{L}_{t-1}\boldsymbol{s} + \boldsymbol{\epsilon}) - \mu\|\boldsymbol{s}\|_1 - \lambda\|\boldsymbol{\epsilon}\|_2^2$. If $\boldsymbol{s} = \boldsymbol{0}$ and $\boldsymbol{\epsilon} = \boldsymbol{0}$, this becomes $\mathcal{J}^{(t)}(\boldsymbol{0})$, which is again bounded, and therefore neither $\boldsymbol{\epsilon}$ nor $\boldsymbol{s}$ may grow too large. The bound on $\boldsymbol{\alpha}^{(t)}$ follows by induction, since $\boldsymbol{\alpha}^{(t)} = \boldsymbol{L}_{t-1}\boldsymbol{s}^{(t)}$. Moreover, since only the $t$−th column of $\boldsymbol{L}$ is modified by setting it to $\boldsymbol{\epsilon}$, $\boldsymbol{L}_t$ is also bounded. For $t > k$,

the same argument applies to $s^{(t)}$ and therefore to $\alpha^{(t)}$. $L_t$ is then given by $\arg\max_L -\lambda\|L\|_F^2 + \frac{1}{t}\sum_{\hat{t}}^t \|Ls^{(\hat{t})} - \alpha^{(\hat{t})}\|_{H^{(\hat{t})}} + g^{(\hat{t})^\top}(Ls^{(\hat{t})} - \alpha^{(\hat{t})})$. If $L_t = 0$, the objective for task $\mathcal{Z}^{(\hat{t})}$ becomes $\alpha^{(\hat{t})^\top} H^{(\hat{t})}\alpha^{(\hat{t})} + g^{(\hat{t})^\top}\alpha^{(\hat{t})}$. By Assumption A and strong induction, this is bounded for all $\hat{t} \leq t$, so if any element of $L$ is too large, $L$ would not be a maximizer because of the regularization term. ∎

**Proposition 1.** $L_t - L_{t-1} = O(\frac{1}{t})$ .

*Proof.* First, we show that $\hat{g}_t - \hat{g}_{t-1}$ is Lipschitz with constant $O\left(\frac{1}{t}\right)$. To show this, we note that $\hat{\ell}$ is Lipschitz in $L$ with a constant independent of $t$, since it is a quadratic function over a compact region with bounded coefficients. Next, we have:

$$\hat{g}_t(L) - \hat{g}_{t-1}(L) = \frac{1}{t}\hat{\ell}\left(L, s^{(t)}, \alpha^{(t)}, H^{(t)}, g^{(t)}\right) + \frac{1}{t}\sum_{\hat{t}=1}^{t-1} \hat{\ell}\left(L, s^{(\hat{t})}, \alpha^{(\hat{t})}, H^{(\hat{t})}, g^{(\hat{t})}\right)$$

$$- \frac{1}{t-1}\sum_{\hat{t}=1}^{t-1} \hat{\ell}\left(L, s^{(\hat{t})}, \alpha^{(\hat{t})}, H^{(\hat{t})}, g^{(\hat{t})}\right)$$

$$= \frac{1}{t}\hat{\ell}\left(L, s^{(t)}, \alpha^{(t)}, H^{(t)}, g^{(t)}\right) + \frac{1}{t(t-1)}\sum_{\hat{t}=1}^{t-1} \hat{\ell}\left(L, s^{(\hat{t})}, \alpha^{(\hat{t})}, H^{(\hat{t})}, g^{(\hat{t})}\right) .$$

Therefore, $\hat{g}_t - \hat{g}_{t-1}$ has a Lipschitz constant $O\left(\frac{1}{t}\right)$, since it is the difference of two terms divided by $t$: $\hat{\ell}$ and an average over $t-1$ terms, whose Lipschitz constant is bounded by the largest Lipschitz constant of the terms.

Let $\xi_t$ be the Lipschitz constant of $\hat{g}_t - \hat{g}_{t-1}$. We have:

$$\hat{g}_{t-1}(L_{t-1}) - \hat{g}_{t-1}(L_t) = \hat{g}_{t-1}(L_{t-1}) - \hat{g}_t(L_{t-1}) + \hat{g}_t(L_{t-1}) - \hat{g}_t(L_t) + \hat{g}_t(L_t) - \hat{g}_{t-1}(L_t)$$

$$\leq \hat{g}_{t-1}(L_{t-1}) - \hat{g}_t(L_{t-1}) + \hat{g}_t(L_t) - \hat{g}_{t-1}(L_t)$$

$$= -(\hat{g}_t - \hat{g}_{t-1})(L_{t-1}) + (\hat{g}_t - \hat{g}_{t-1})(L_t) \leq \xi_t\|L_t - L_{t-1}\|_F .$$

Moreover, since $L_{t-1}$ maximizes $\hat{g}_{t-1}$ and the $\ell_2$ regularization term ensures that the maximum eigenvalue of the Hessian of $\hat{g}_{t-1}$ is upper-bounded by $-2\lambda$, we have that $\hat{g}_{t-1}(L_{t-1}) - \hat{g}_{t-1}(L_t) \geq \lambda\|L_t - L_{t-1}\|_F^2$. Combining these two inequalities, we have: $\|L_t - L_{t-1}\|_F \leq \frac{\xi_t}{\lambda} = O\left(\frac{1}{t}\right)$. ∎

The critical step for adapting the proof from Ruvolo and Eaton [32] to LPG-FTW is to introduce the following lemma, which shows the equality of the maximizers of $\ell$ and $\hat{\ell}$.

**Lemma 1.** $\hat{\ell}\left(L_t, s^{(t+1)}, \alpha^{(t+1)}, H^{(t+1)}, g^{(t+1)}\right) = \max_s \hat{\ell}\left(L_t, s, \alpha^{(t+1)}, H^{(t+1)}, g^{(t+1)}\right)$ .

*Proof.* To show this, we need the following to hold:

$$s^{(t+1)} = \arg\max_s \ell(L_t, s) = \arg\max_s \hat{\ell}\left(L_t, s, \alpha^{(t+1)}, H^{(t+1)}, g^{(t+1)}\right) .$$

We first compute the gradient of $\ell$, given by:

$$\nabla_s \ell(L_t, s) = -\mu\,\mathrm{sign}(s) + L_t^\top \nabla_\theta \mathcal{J}^{(t+1)}(\theta)\Big|_{\theta = L_t s} .$$

Since $s^{(t+1)}$ is the maximizer of $\ell$, we have:

$$\nabla_s \ell(L_t, s)\Big|_{s=s^{(t+1)}} = -\mu\,\mathrm{sign}\left(s^{(t+1)}\right) + L_t^\top g^{(t+1)} = 0 . \tag{A.1}$$

We now compute the gradient of $\hat{\ell}$ and evaluate it at $s^{(t+1)}$:

$$\nabla_s \hat{\ell}\left(L_t, s, \alpha^{(t+1)}, H^{(t+1)}, g^{(t+1)}\right) = -\mu\,\mathrm{sign}\left(s^{(t+1)}\right) + L_t g^{(t+1)} - 2L_t^\top H^{(t+1)}\left(\alpha^{(t+1)} - L_t s\right)$$

$$\nabla_s \hat{\ell}\left(L_t, s, \alpha^{(t+1)}, H^{(t+1)}, g^{(t+1)}\right)\Big|_{s=s^{(t+1)}} = -\mu\,\mathrm{sign}\left(s^{(t+1)}\right) + L_t^\top g^{(t+1)} = 0 ,$$

since it matches Equation A.1. By Assumption D, $\hat{\ell}$ has a unique maximizer $s^{(t+1)}$. ∎

Before stating our next lemma, we define:

$$s^* = \beta\left(\boldsymbol{L}, \boldsymbol{\alpha}^{(t)}, \boldsymbol{H}^{(t)}, \boldsymbol{g}^{(t)}\right) = \arg\max_s \hat{\ell}\left(\boldsymbol{L}, \boldsymbol{s}, \boldsymbol{\alpha}^{(t)}, \boldsymbol{H}^{(t)}, \boldsymbol{g}^{(t)}\right) \ .$$

**Lemma 2.**

    A. $\max_s \hat{\ell}(\boldsymbol{L}, \boldsymbol{s}, \boldsymbol{\alpha}^{(t)}, \boldsymbol{H}^{(t)}, \boldsymbol{g}^{(t)})$     *is*     *continuously*     *differentiable*    *in*    $\boldsymbol{L}$    *with*
        $\nabla_{\boldsymbol{L}} \max_s \hat{\ell}(\boldsymbol{L}, \boldsymbol{s}, \boldsymbol{\alpha}^{(t)}, \boldsymbol{H}^{(t)}, \boldsymbol{g}^{(t)}) = \left[-2\boldsymbol{H}^{(t)}\boldsymbol{s}^* + \boldsymbol{g}^{(t)}\right]\boldsymbol{s}^{*\top}.$

    B.  *$g$ is continuously differentiable and* $\nabla_{\boldsymbol{L}} g(\boldsymbol{L}) = -2\lambda\boldsymbol{I} + \mathbb{E}\left[\nabla_{\boldsymbol{L}} \max_s \hat{\ell}(\boldsymbol{L}, \boldsymbol{s}, \boldsymbol{\alpha}^{(t)}, \boldsymbol{H}^{(t)}, \boldsymbol{g}^{(t)})\right].$

    C.  $\nabla_{\boldsymbol{L}} g(\boldsymbol{L})$ *is Lipschitz in the space of latent components $\boldsymbol{L}$ that obey Claim 1.*

*Proof.* To prove Part A, we apply a corollary to Theorem 4.1 in [3]. This corollary states that if $\hat{\ell}$ is continuously differentiable in $\boldsymbol{L}$ (which it clearly is) and has a unique maximizer $\boldsymbol{s}^{(t)}$ (which is guaranteed by Assumption D), then $\nabla_{\boldsymbol{L}} \min_s \hat{\ell}(\boldsymbol{L}, \boldsymbol{s}, \boldsymbol{\alpha}^{(t)}, \boldsymbol{H}^{(t)}, \boldsymbol{g}^{(t)})$ exists and is equal to $\nabla_{\boldsymbol{L}}\hat{\ell}(\boldsymbol{L}, \boldsymbol{s}^*, \boldsymbol{\alpha}^{(t)}, \boldsymbol{H}^{(t)}, \boldsymbol{g}^{(t)})$, given by $\left[-2\boldsymbol{H}^{(t)}\boldsymbol{s}^* + \boldsymbol{g}^{(t)}\right]\boldsymbol{s}^{*\top}$. Part B follows since by Assumption A and Claim 1 the tuple $\left(\boldsymbol{H}^{(t)}, \boldsymbol{g}^{(t)}, \boldsymbol{\alpha}^{(t)}\right)$ is drawn from a distribution with compact support.

To prove Part C, we first show that $\beta$ is Lipschitz in $\boldsymbol{L}$ with constant independent of $\boldsymbol{\alpha}^{(t)}$, $\boldsymbol{H}^{(t)}$, and $\boldsymbol{g}^{(t)}$. Part C will follow due to the form of the gradient of $g$ with respect to $\boldsymbol{L}$. The function $\beta$ is continuous in its arguments since $\hat{\ell}$ is continuous in its arguments and by Assumption D has a unique maximizer. Next, we define $\rho\left(\boldsymbol{L}, \boldsymbol{H}^{(t)}, \boldsymbol{g}^{(t)}, \boldsymbol{\alpha}^{(t)}, j\right) = \boldsymbol{l}_j^\top\left[2\boldsymbol{H}^{(t)}\left(\boldsymbol{L}\boldsymbol{s}^* - \boldsymbol{\alpha}^{(t)}\right) + \boldsymbol{g}^{(t)}\right]$, where $\boldsymbol{l}_j$ is the $j$-th column of $\boldsymbol{L}$. Following the argument of Fuchs [13], we reach the following conditions:

$$\left|\rho\left(\boldsymbol{L}, \boldsymbol{H}^{(t)}, \boldsymbol{g}^{(t)}, \boldsymbol{\alpha}^{(t)}, j\right)\right| = \mu \iff \boldsymbol{s}_j^* \neq 0$$
$$\left|\rho\left(\boldsymbol{L}, \boldsymbol{H}^{(t)}, \boldsymbol{g}^{(t)}, \boldsymbol{\alpha}^{(t)}, j\right)\right| < \mu \iff \boldsymbol{s}_j^* = 0 \ . \tag{A.2}$$

Let $\gamma$ be the set of indices $j$ such that $\left|\rho\left(\boldsymbol{L}, \boldsymbol{H}^{(t)}, \boldsymbol{g}^{(t)}, \boldsymbol{\alpha}^{(t)}, j\right)\right| = \mu$. Since $\rho$ is continuous in $\boldsymbol{L}$, $\boldsymbol{H}^{(t)}$, $\boldsymbol{g}^{(t)}$, and $\boldsymbol{\alpha}^{(t)}$, there must exist an open neighborhood $V$ around $\left(\boldsymbol{L}, \boldsymbol{H}^{(t)}, \boldsymbol{g}^{(t)}, \boldsymbol{\alpha}^{(t)}\right)$ such that for all $\left(\boldsymbol{L}', \boldsymbol{H}^{(t)'}, \boldsymbol{g}^{(t)'}, \boldsymbol{\alpha}^{(t)'}\right) \in V$ and $j \notin \gamma$, $\left|\rho\left(\boldsymbol{L}', \boldsymbol{H}^{(t)'}, \boldsymbol{g}^{(t)'}, \boldsymbol{\alpha}^{(t)'}, j\right)\right| < \mu$. By Equation A.2, we conclude that $\beta\left(\boldsymbol{L}', \boldsymbol{H}^{(t)'}, \boldsymbol{g}^{(t)'}, \boldsymbol{\alpha}^{(t)'}\right)_j = 0, \forall j \notin \gamma$.

Next, we define a new objective:

$$\bar{\ell}(\boldsymbol{L}_\gamma, \boldsymbol{s}_\gamma, \boldsymbol{\alpha}, \boldsymbol{H}, \boldsymbol{g}) = \|\boldsymbol{\alpha} - \boldsymbol{L}_\gamma\boldsymbol{s}_\gamma\|_{\boldsymbol{H}}^2 + \boldsymbol{g}^\top(\boldsymbol{L}_\gamma\boldsymbol{s}_\gamma - \boldsymbol{\alpha}) - \mu\|\boldsymbol{s}_\gamma\|_1 \ .$$

By Assumption D, $\bar{\ell}$ is strictly concave with a Hessian upper-bounded by $-2\kappa$. We can conclude that:

$$\bar{\ell}\left(\boldsymbol{L}_\gamma, \beta\left(\boldsymbol{L}, \boldsymbol{\alpha}^{(t)}, \boldsymbol{H}^{(t)}, \boldsymbol{g}^{(t)}\right)_\gamma, \boldsymbol{\alpha}^{(t)}, \boldsymbol{H}^{(t)}, \boldsymbol{g}^{(t)}\right) - \bar{\ell}\left(\boldsymbol{L}_\gamma, \beta\left(\boldsymbol{L}', \boldsymbol{\alpha}^{(t)'}, \boldsymbol{H}^{(t)'}, \boldsymbol{g}^{(t)'}\right)_\gamma, \boldsymbol{\alpha}^{(t)}, \boldsymbol{H}^{(t)}, \boldsymbol{g}^{(t)}\right)$$

$$\geq \kappa \left\|\beta\left(\boldsymbol{L}', \boldsymbol{\alpha}^{(t)'}, \boldsymbol{H}^{(t)'}, \boldsymbol{g}^{(t)'}\right)_\gamma - \beta\left(\boldsymbol{L}, \boldsymbol{\alpha}^{(t)}, \boldsymbol{H}^{(t)}, \boldsymbol{g}^{(t)}\right)_\gamma\right\|_2^2 \ . \tag{A.3}$$

On the other hand, by Assumption A and Claim 1, $\bar{\ell}$ is Lipschitz in its second argument with constant $e_1\|\boldsymbol{L}_\gamma - \boldsymbol{L}_\gamma'\|_{\mathsf{F}} + e_2\|\boldsymbol{\alpha} - \boldsymbol{\alpha}'\|_2 + e_3\|\boldsymbol{H} - \boldsymbol{H}'\|_{\mathsf{F}} + e_4\|\boldsymbol{g} - \boldsymbol{g}'\|_2$, where $e_{1-4}$ are all constants independent of any of the arguments. Combining this with Equation A.3, we obtain:

$$\left\|\beta\left(\boldsymbol{L}', \boldsymbol{\alpha}^{(t)'}, \boldsymbol{H}^{(t)'}, \boldsymbol{g}^{(t)'}\right) - \beta\left(\boldsymbol{L}, \boldsymbol{\alpha}^{(t)}, \boldsymbol{H}^{(t)}, \boldsymbol{g}^{(t)}\right)\right\| =$$

$$\left\|\beta\left(\boldsymbol{L}', \boldsymbol{\alpha}^{(t)'}, \boldsymbol{H}^{(t)'}, \boldsymbol{g}^{(t)'}\right)_\gamma - \beta\left(\boldsymbol{L}, \boldsymbol{\alpha}^{(t)}, \boldsymbol{H}^{(t)}, \boldsymbol{g}^{(t)}\right)_\gamma\right\|$$

$$\leq \frac{e_1\|\boldsymbol{L}_\gamma - \boldsymbol{L}_\gamma'\|_{\mathsf{F}} + e_2\|\boldsymbol{\alpha}^{(t)} - \boldsymbol{\alpha}^{(t)'}\|_2}{\kappa} + \frac{e_3\|\boldsymbol{H}^{(t)} - \boldsymbol{H}^{(t)'}\|_{\mathsf{F}} + e_4\|\boldsymbol{g}^{(t)} - \boldsymbol{g}^{(t)'}\|_2}{\kappa} \ .$$

Therefore, $\beta$ is locally Lipschitz. Since the domain of $\beta$ is compact by Assumption A and Claim 1, this implies that $\beta$ is uniformly Lipschitz, and we can conclude that $\nabla g$ is Lipschitz as well. ∎

**Proposition 2.**  1. $\hat{g}_t(\boldsymbol{L}_t)$ *converges a.s.*  3. $g_t(\boldsymbol{L}_t) - \hat{g}(\boldsymbol{L}_t)$ *converges a.s. to 0*
  2. $g_t(\boldsymbol{L}_t) - \hat{g}_t(\boldsymbol{L}_t)$ *converges a.s. to 0*  4. $g(\boldsymbol{L}_t)$ *converges a.s.*

*Proof.* We begin by defining the stochastic process $u_t = \hat{g}_t(\boldsymbol{L})$. The outline of the proof is to show that this process is a quasi-martingale and by a theorem by Fisk [12], it converges almost surely.

$$
\begin{aligned}
u_{t+1} - u_t =& \hat{g}_{t+1}(\boldsymbol{L}_{t+1}) - \hat{g}_t(\boldsymbol{L}_t) = \hat{g}_{t+1}(\boldsymbol{L}_{t+1}) - \hat{g}_{t+1}(\boldsymbol{L}_t) + \hat{g}_{t+1}(\boldsymbol{L}_t) - \hat{g}_t(\boldsymbol{L}_t) \\
=& (\hat{g}_{t+1}(\boldsymbol{L}_{t+1}) - \hat{g}_{t+1}(\boldsymbol{L}_t)) + \frac{g_t(\boldsymbol{L}_t) - \hat{g}_t(\boldsymbol{L}_t)}{t+1} \\
& + \frac{\max_{\boldsymbol{s}} \hat{\ell}(\boldsymbol{L}_t, \boldsymbol{s}, \boldsymbol{\alpha}^{(t+1)}, \boldsymbol{H}^{(t+1)}, \boldsymbol{g}^{(t+1)})}{t+1} - \frac{g_t(\boldsymbol{L}_t)}{t+1} \quad ,
\end{aligned}
\tag{A.4}
$$

where we made use of the fact that:

$$
\begin{aligned}
\hat{g}_{t+1}(\boldsymbol{L}_t) =& \frac{\hat{\ell}\Big(\boldsymbol{L}_t, \boldsymbol{s}^{(t+1)}, \boldsymbol{\alpha}^{(t+1)}, \boldsymbol{H}^{(t+1)}, \boldsymbol{g}^{(t+1)}\Big)}{t+1} + \frac{t}{t+1}\hat{g}_t(\boldsymbol{L}_t) \\
=& \frac{\max_{\boldsymbol{s}} \hat{\ell}\Big(\boldsymbol{L}_t, \boldsymbol{s}, \boldsymbol{\alpha}^{(t+1)}, \boldsymbol{H}^{(t+1)}, \boldsymbol{g}^{(t+1)}\Big)}{t+1} + \frac{t}{t+1}\hat{g}_t(\boldsymbol{L}_t) \quad ,
\end{aligned}
$$

where the second equality holds by Lemma 1.

We now need to show that the sum of positive and negative variations in Equation A.4 are bounded. By an argument similar to a lemma by Bottou [4], the sum of positive variations of $u_t$ is bounded, since $\hat{g}$ is upper-bounded by Assumption C. Therefore, it suffices to show that the sum of negative variations is bounded. The first term on the first line of Equation A.4 is guaranteed to be positive since $\boldsymbol{L}_{t+1}$ maximizes $\hat{g}_{t+1}$. Additionally, since $g_t$ is always at least as large as $\hat{g}_t$, the second term on the first line is also guaranteed to be positive. Therefore, we focus on the second line.

$$
\begin{aligned}
\mathbb{E}[u_{t+1} - u_t \mid \mathcal{I}_t] \geq& \frac{\mathbb{E}\Big[\max_{\boldsymbol{s}} \hat{\ell}\Big(\boldsymbol{L}_t, \boldsymbol{s}, \boldsymbol{\alpha}^{(t+1)}, \boldsymbol{H}^{(t+1)}, \boldsymbol{g}^{(t+1)}\Big) \mid \mathcal{I}_t\Big]}{t+1} - \frac{g_t(\boldsymbol{L}_t)}{t+1} \\
=& \frac{g(\boldsymbol{L}_t) - g_t(\boldsymbol{L}_t)}{t+1} \geq -\frac{\|g - g_t\|_\infty}{t+1} \quad ,
\end{aligned}
$$

where $\mathcal{I}_t$ represents all the $\boldsymbol{\alpha}^{(\hat{t})}$'s, $\boldsymbol{H}^{(\hat{t})}$'s, and $\boldsymbol{g}^{(\hat{t})}$'s up to time $t$. Hence, showing that $\sum_{t=1}^{\infty} \frac{\|g - g_t\|_\infty}{t+1} < \infty$ will prove that $u_t$ is a quasi-martingale that converges almost surely.

In order to prove this, we apply the following corollary of the Donsker theorem [39]:

Let $\mathcal{F} = \{f_{\boldsymbol{\theta}} : \mathcal{X} \mapsto \mathbb{R}, \boldsymbol{\theta} \in \boldsymbol{\Theta}\}$ be a set of measurable functions indexed by a bounded subset $\boldsymbol{\Theta}$ of $\mathbb{R}^d$. Suppose that there exists a constant $K$ such that:

$$
|f_{\boldsymbol{\theta}_1}(x) - f_{\boldsymbol{\theta}_2}(x)| \leq K\|\boldsymbol{\theta}_1 - \boldsymbol{\theta}_2\|_2
$$

for every $\boldsymbol{\theta}_1, \boldsymbol{\theta}_2 \in \boldsymbol{\Theta}$ and $x \in \mathcal{X}$. Then, $\mathcal{F}$ is P-Donsker and for any $f \in \mathcal{F}$, we define:

$$
\begin{aligned}
\mathbb{P}_n f =& \frac{1}{n} \sum_{i=1}^{n} f(X_i) \\
\mathbb{P}f =& \mathbb{E}_X[f(X)] \\
\mathbb{G}_n f =& \sqrt{n}(\mathbb{P}_n f - \mathbb{P}f) \quad .
\end{aligned}
$$

If $\mathbb{P}f^2 \leq \delta^2$ and $\|f\|_\infty < B$ and the random elements are Borel measurable, then:

$$
\mathbb{E}[\sup_{f \in \mathcal{F}} |\mathbb{G}_n f|] = O(1) \quad .
$$

In order to apply this corollary to our analysis, consider a set of functions $\mathcal{F}$ indexed by $\boldsymbol{L}$, given by $f_{\boldsymbol{L}}\big(\boldsymbol{H}^{(t)}, \boldsymbol{g}^{(t)}, \boldsymbol{\alpha}^{(t)}\big) = \max_{\boldsymbol{s}} \hat{\ell}(\boldsymbol{L}, \boldsymbol{s}, \boldsymbol{\alpha}^{(t)}, \boldsymbol{H}^{(t)}, \boldsymbol{g}^{(t)})$, whose domain is all possible tuples $\big(\boldsymbol{H}^{(t)}, \boldsymbol{g}^{(t)}, \boldsymbol{\alpha}^{(t)}\big)$. The expected value of $f^2$ is bounded for all $f \in \mathcal{F}$ since $\hat{\ell}$ is bounded by

Claim 1. Second, $\|f\|_\infty$ is bounded given Claim 1 and Assumption A. Finally, by Assumptions A and B, the corollary applies to the tuples $(\boldsymbol{H}^{(t)}, \boldsymbol{g}^{(t)}, \boldsymbol{\alpha}^{(t)})$ [2]. Therefore, we can state that:

$$\mathbb{E}\left[\sqrt{t}\left\|\left(\frac{1}{t}\sum_{\hat{t}=1}^{t}\max_{\boldsymbol{s}}\hat{\ell}\left(\boldsymbol{L}, \boldsymbol{s}, \boldsymbol{\alpha}^{(\hat{t})}, \boldsymbol{H}^{(\hat{t})}, \boldsymbol{g}^{(\hat{t})}\right)\right) - \mathbb{E}\left[\max_{\boldsymbol{s}}\hat{\ell}\left(\boldsymbol{L}, \boldsymbol{s}, \boldsymbol{\alpha}^{(\hat{t})}, \boldsymbol{H}^{(\hat{t})}, \boldsymbol{g}^{(\hat{t})}\right)\right]\right\|_\infty\right] = O(1)$$

$$\implies \mathbb{E}[\|g_t(\boldsymbol{L}) - g(\boldsymbol{L})\|_\infty] = O\left(\frac{1}{\sqrt{t}}\right) \ .$$

Therefore, $\exists\, c_3 \in \mathbb{R}$ such that $\mathbb{E}[\|g_t - g\|_\infty] < \frac{c_3}{\sqrt{t}}$:

$$\sum_{t=1}^{\infty}\mathbb{E}\left[\mathbb{E}[u_{t+1} - u_t \mid \mathcal{I}_t]^-\right] \geq \sum_{t=1}^{\infty} -\frac{\mathbb{E}[\|g_t - g\|_\infty]}{t+1} > \sum_{t=1}^{\infty} -\frac{c_3}{t^{\frac{3}{2}}} = -O(1) \ ,$$

where $i^- = \min(i, 0)$. This shows that the sum of negative variations of $u_t$ is bounded, so $u_t$ is a quasi-martingale and thus converges almost surely [12]. This proves Part 1 of Proposition 2.

Next, we show that $u_t$ being a quasi-martingale implies the almost sure convergence of the fourth line of Equation A.4. To see this, we note that since $u_t$ is a quasi-martingale and the sum of its positive variations is bounded, and since the term on the fourth line of Equation A.4, $\frac{g_t(\boldsymbol{L}_t) - \hat{g}_t(\boldsymbol{L}_t)}{t+1}$, is guaranteed to be positive, the sum of that term from 1 to infinity must be bounded:

$$\sum_{t=1}^{\infty}\frac{g_t(\boldsymbol{L}_t) - \hat{g}_t(\boldsymbol{L}_t)}{t+1} < \infty \ . \tag{A.5}$$

To complete the proof of Part 2 of Proposition 2, consider the following lemma: Let $a_n, b_n$ be two real sequences such that for all $n$, $a_n \geq 0, b_n \geq 0, \sum_{j=1}^{\infty} a_j = \infty, \sum_{j=1}^{\infty} a_j b_j < \infty, \exists\, K > 0$ such that $|b_{n+1} - b_n| < Ka_n$. Then, $\lim_{n\to\infty} b_n = 0$. If we define $a_t = \frac{1}{t+1}$ and $b_t = g_t(\boldsymbol{L}_t) - \hat{g}_t(\boldsymbol{L}_t)$, then clearly these are both positive sequences and $\sum_{t=1}^{\infty} a_t = \infty$. By Equation A.5, $\sum_{t=1}^{\infty} a_n b_n < \infty$. Finally, since $g_t$ and $\hat{g}_t$ are bounded and Lipschitz with constant independent of $t$ and $\boldsymbol{L}_{t+1} - \boldsymbol{L}_t = O\left(\frac{1}{t}\right)$, we have all of the assumptions verified, which implies that $g_t - \hat{g}_t$ converges a.s. to 0.

By Part 2 and the Glivenko-Cantelli theorem, $\lim_{t\to\infty} \|g - g_t\|_\infty = 0$, which implies that $g$ must converge almost surely. By transitivity, $\lim_{t\to\infty} g(\boldsymbol{L}_t) - \hat{g}_t(\boldsymbol{L}_t) = 0$, showing Parts 3 and 4. ∎

**Proposition 3.** *The distance between $\boldsymbol{L}_t$ and the set of all stationary points of $g$ converges a.s. to 0.*

*Proof.* First, $\nabla_{\boldsymbol{L}}\hat{g}_t$ is Lipschitz with a constant independent of $t$, since the gradient of $\hat{g}_t$ is linear, $\boldsymbol{s}^{(t)}, \boldsymbol{H}^{(t)}, \boldsymbol{g}^{(t)}$, and $\boldsymbol{\alpha}^{(t)}$ are bounded, and the summation in $\hat{g}_t$ is normalized by $t$. Next, we define an arbitrary non-zero matrix $\boldsymbol{U}$ of the same dimensionality as $\boldsymbol{L}$. Since $g_t$ upper-bounds $\hat{g}_t$, we have:

$$g_t(\boldsymbol{L}_t + \boldsymbol{U}) \geq \hat{g}_t(\boldsymbol{L}_t + \boldsymbol{U}) \implies \lim_{t\to\infty} g(\boldsymbol{L}_t + \boldsymbol{U}) \geq \lim_{t\to\infty} \hat{g}_t(\boldsymbol{L}_t + \boldsymbol{U}) \ ,$$

where we used the fact that $\lim_{t\to\infty} g_t = \lim_{t\to\infty} g$. Let $h_t > 0$ be a sequence of positive real numbers that converges to 0. If we take the first-order Taylor expansion on both sides of the inequality and use the fact that $\nabla g$ and $\nabla \hat{g}$ are both Lipschitz with constant independent of $t$, we get:

$$\lim_{t\to\infty} g_t(\boldsymbol{L}_t) + \text{Tr}\left(h_t\boldsymbol{U}^\top\nabla g_t(\boldsymbol{L}_t)\right) + O(h_t\boldsymbol{U}) \geq \lim_{t\to\infty} \hat{g}_t(\boldsymbol{L}_t) + \text{Tr}\left(h_t\boldsymbol{U}^\top\nabla\hat{g}_t(\boldsymbol{L}_t)\right) + O(h_t\boldsymbol{U}) \ .$$

Since $\lim_{t\to\infty} g(\boldsymbol{L}_t) - \hat{g}(\boldsymbol{L}_t) = 0$ a.s. and $\lim_{t\to\infty} h_t = 0$, we have:

$$\lim_{t\to\infty}\left(\frac{1}{\|\boldsymbol{U}\|_{\mathsf{F}}}\boldsymbol{U}^\top\nabla g(\boldsymbol{L}_t)\right) \geq \lim_{t\to\infty}\left(\frac{1}{\|\boldsymbol{U}\|_{\mathsf{F}}}\boldsymbol{U}^\top\nabla\hat{g}(\boldsymbol{L}_t)\right) \ .$$

Since this inequality has to hold for every $\boldsymbol{U}$, we require that $\lim_{t\to\infty}\nabla g(\boldsymbol{L}_t) = \lim_{t\to\infty}\nabla\hat{g}_t(\boldsymbol{L}_t)$. Since $\boldsymbol{L}_t$ minimizes $\hat{g}_t$, we require that $\nabla\hat{g}_t(\boldsymbol{L}_t) = \boldsymbol{0}$. This implies that $\nabla g(\boldsymbol{L}_t) = \boldsymbol{0}$, which is a sufficient first-order condition for $\boldsymbol{L}_t$ to be stationary point of $g$. ∎

## B  Experimental setting

This section provides additional details of the experimental setting used to arrive at the results presented in Sections 6.1 and 6.2 in the main paper. Table B.1 summarizes the hyper-parameters of all algorithms used for our experiments.

Table B.1: Summary of hyper-parameters. The first digit in EWC versions differentiates variants with shared $\sigma$ (1) and task-specific $\sigma$ (2), and the second digit differentiates between Huszár regularization (1), EWC regularization scaled by $\frac{1}{t-1}$ (2), and the original EWC regularization (3). The boldfaced version of EWC was used for our experiments in the paper.

|  | Hyper-parameter | HC-G | HC-BP | Ho-G | Ho-BP | W-G | W-B | MT10/50 |
|---|---|---|---|---|---|---|---|---|
| NPG | # iterations | 50 | 50 | 100 | 100 | 200 | 200 | 200 |
|  | # trajectories / iter | 10 | 10 | 50 | 50 | 50 | 50 | 50 |
|  | step size | 0.5 | 0.5 | 0.005 | 0.005 | 0.05 | 0.05 | 0.005 |
|  | $\lambda$ (GAE) | 0.97 | 0.97 | 0.97 | 0.97 | 0.97 | 0.97 | 0.97 |
|  | $\gamma$ (MDP) | 0.995 | 0.995 | 0.995 | 0.995 | 0.995 | 0.995 | 0.995 |
| LPG-FTW | $\lambda$ | $1e{-}5$ | $1e{-}5$ | $1e{-}5$ | $1e{-}5$ | $1e{-}5$ | $1e{-}5$ | $1e{-}5$ |
|  | $\mu$ | $1e{-}5$ | $1e{-}5$ | $1e{-}5$ | $1e{-}5$ | $1e{-}5$ | $1e{-}5$ | $1e{-}5$ |
|  | $k$ | 5 | 5 | 5 | 5 | 5 | 10 | 3 |
| PG-ELLA | $\lambda$ | $1e{-}5$ | $1e{-}5$ | $1e{-}5$ | $1e{-}5$ | $1e{-}5$ | $1e{-}5$ | $1e{-}5$ |
|  | $\mu$ | $1e{-}5$ | $1e{-}5$ | $1e{-}5$ | $1e{-}5$ | $1e{-}5$ | $1e{-}5$ | $1e{-}5$ |
|  | $k$ | 5 | 5 | 5 | 5 | 5 | 10 | 3 |
| EWC 1, 1 | $\lambda$ | $1e{-}3$ | $1e{-}6$ | $1e{-}7$ | $1e{-}7$ | $1e{-}4$ | $1e{-}5$ | — |
| EWC 1, 2 | $\lambda$ | $1e{-}3$ | $1e{-}4$ | $1e{-}7$ | $1e{-}3$ | $1e{-}3$ | $1e{-}7$ | — |
| **EWC 1, 3** | $\lambda$ | $1e{-}6$ | $1e{-}6$ | $1e{-}7$ | $1e{-}4$ | $1e{-}7$ | $1e{-}7$ | — |
| EWC 2, 1 | $\lambda$ | $1e{-}6$ | $1e{-}5$ | $1e{-}6$ | $1e{-}7$ | $1e{-}4$ | $1e{-}4$ | — |
| EWC 2, 2 | $\lambda$ | $1e{-}4$ | $1e{-}3$ | $1e{-}5$ | $1e{-}7$ | $1e{-}4$ | $1e{-}6$ | — |
| EWC 2, 3 | $\lambda$ | $1e{-}4$ | $1e{-}4$ | $1e{-}6$ | $1e{-}7$ | $1e{-}4$ | $1e{-}4$ | $1e{-}7$ |

**OpenAI Gym MuJoCo domains** The hyper-parameters for NPG were manually selected by running an evaluation on the nominal task for each domain (without gravity or body part modifications). We tried various combinations of the number of iterations, number of trajectories per iteration, and step size, until we reached a learning curve that was fast and reached proficiency. Once these hyper-parameters were found, they were used for all lifelong learning algorithms. For LPG-FTW, we chose typical hyper-parameters and held them fixed through all experiments, forgoing potential additional benefits from a hyper-parameter search. The only exception was the number of latent components used for the Walker-2D *body-parts* domain, as we found empirically that $k = 5$ led to saturation of the learning process early on. For PG-ELLA, we kept the same hyper-parameters as used for LPG-FTW, since they are used in exactly the same way for both methods. Finally, for EWC, we ran a grid search over the value of the regularization term, $\lambda$, among $\{1e{-}7, 1e{-}6, 1e{-}5, 1e{-}4, 1e{-}3\}$. The search was done by running five consecutive tasks for fifty iterations over five trials with different random seeds. We chose $\lambda$ independently for each domain to maximize the average performance after all tasks had been trained. We also tried various versions of EWC, as described in Appendix D, modifying the regularization term and selecting whether to share the policy's variance across tasks. The only version that worked in all domains was the original EWC penalty with a shared variance across tasks, so results in the main paper are based on that version. To make comparisons fair, EWC used the full Hessian instead of the diagonal Hessian proposed by the authors.

**Meta-World domains** In this case, we manually tuned the hyper-parameters for NPG on the `reach` task, which we considered to be the easiest to solve in the benchmark, and again kept those fixed for all lifelong learners. We chose typical values for LPG-FTW for $k$, $\lambda$, and $\mu$, and reused those for PG-ELLA. We used fewer latent components ($k = 3$), since MT10 contains only $T_{\mathsf{max}} = 10$ tasks and we considered that using more than three policy factors would give our algorithm an unfair advantage over single-model methods. For EWC and EWC_h, we ran a grid search for $\lambda$ in the same way as for the previous experiments. For ER, we used a 50-50 ratio of experience replay as suggested by Rolnick et al. [30], and ensured that each batch sampled from the replay buffer had the same number of trajectories from each previous task. LPG-FTW, PG-ELLA, and EWC all had access to the full Hessian, and we chose for EWC not to share the variance across tasks since the outputs of the policies were task-specific. We ran all Meta-World tasks on version 1.5 of the MuJoCo physics simulator, to match the remainder of our experimental setting [38]. We used the robot hand and the object location (6-D) as the observation space for all tasks. Note that the goal, which was kept fixed for each task, was not given to the agent. For this reason, we removed 2 tasks from MT50 that use at least 9-D observations—`stick pull` and `stick push`—for a total of $T_{\mathsf{max}} = 48$ tasks.

## C Diversity of simple domains

One important question in the study of lifelong RL is how diverse the tasks used for evaluation are. To measure this in the OpenAI Gym MuJoCo domains, we evaluated each task's performance using the final policy trained by LPG-FTW on the correct task and compared it to the average performance using the policies trained on all other tasks. Figure C.1 shows that the policies do not work well across different tasks, demonstrating that the tasks are diverse. Moreover, the most highly-varying domains, Hopper and Walker-2D *body-parts*, are precisely those for which EWC struggled the most, suffering from catastrophic forgetting, as shown in Figure 2 in the paper. This is consistent with the fact that a single policy does not work across various tasks. In those domains, LPG-FTW reached the performance of STL with a high speedup while retaining knowledge from early tasks.

## D EWC additional results

While testing on OpenAI MuJoCo domains, we experimented with six different variants of EWC, by varying two different choices. The first choice was whether to share the variance of the Gaussian policies across the different tasks or not. Sharing the variance enables the algorithm to start from a more deterministic policy, thereby achieving higher initial performance, at the cost of reducing task-specific exploration. The second choice was the exact form of the regularization penalty. In the original EWC formulation, the regularization term applied to the PG objective was $-\lambda \sum_{\hat{t}=1}^{t-1} \|\boldsymbol{\theta} - \boldsymbol{\alpha}^{(\hat{t})}\|_{\boldsymbol{H}^{(\hat{t})}}^2$. Huszár [17] noted that this does not correspond to the correct Bayesian formulation, and proposed to instead use $-\lambda \|\boldsymbol{\theta} - \boldsymbol{\alpha}^{(t-1)}\|_{\boldsymbol{H}^{(t-1)}}^2$, where $\boldsymbol{\alpha}^{(t-1)}$ and $\boldsymbol{H}^{(t-1)}$ capture all the information from tasks 1 through $t-1$ in the Bayesian setting. We experimented with these two choices of regularization, plus an additional one where $\lambda$ is scaled by $\frac{1}{t-1}$ in order for the penalty not to increase linearly with the number of tasks. For all versions, we independently tuned the hyper-parameters as described in Appendix B.

Table D.1 summarizes the results obtained for each variant of EWC. The only version that consistently learned each task's policy (tune) was the original EWC regularization with the variance shared across tasks. This was also the only variant for which the final performance was never unreasonably low. Therefore, we used this version for all experiments in the main paper.

Figure C.1: Performance with the true policy vs other policies. Percent gap ($\Delta$) indicates task diversity. Body parts (BP) domains are more diverse than gravity (G) domains, and Walker-2D (W) and Hopper (Ho) domains are more varied than HalfCheetah (HC) domains.

Table D.1: Results with different versions of EWC. EWC regularization with $\sigma$ shared across tasks (boldfaced) was the most consistent, so we chose this for our experiments in the main paper. NaN's indicate that the learned policies became unstable, leading to failures in the simulator.

| Domain | Algorithm | Start | Tune | Final |
|---|---|---|---|---|
| HC_G | EWC 1, 1 | −1245 | −2917 | −3.97e4 |
| | EWC 1, 2 | 1796 | 2409 | 2603 |
| | **EWC 1, 3** | **1666** | **2225** | **2254** |
| | EWC 2, 1 | −778 | 1384 | 1797 |
| | EWC 2, 2 | −762 | 1565 | 2238 |
| | EWC 2, 3 | −6.58e5 | −7e5 | −1.05e7 |
| HC_BP | EWC 1, 1 | 1029 | 1748 | 1522 |
| | EWC 1, 2 | 1132 | 1769 | 1588 |
| | **EWC 1, 3** | **1077** | **1716** | **1571** |
| | EWC 2, 1 | −892 | 1308 | 1521 |
| | EWC 2, 2 | −1.79e5 | −2.16e5 | −1.53e6 |
| | EWC 2, 3 | −1.1e6 | −1.01e6 | −5.23e6 |
| Ho_G | EWC 1, 1 | 1301 | 2252 | 1522 |
| | EWC 1, 2 | 1339 | 2322 | 1836 |
| | **EWC 1, 3** | **1434** | **2488** | **1732** |
| | EWC 2, 1 | 872 | 2616 | 2089 |
| | EWC 2, 2 | 930 | 2582 | 1900 |
| | EWC 2, 3 | 939 | 2520 | 2029 |
| Ho_BP | EWC 1, 1 | 613 | 1508 | 793 |
| | EWC 1, 2 | 385 | 920 | 43 |
| | **EWC 1, 3** | **424** | **936** | **31** |
| | EWC 2, 1 | 615 | 2142 | 1011 |
| | EWC 2, 2 | 620 | 2119 | 1120 |
| | EWC 2, 3 | 613 | 2138 | 928 |
| W_G | EWC 1, 1 | 1293 | 2052 | 303 |
| | EWC 1, 2 | −2132 | −2181 | NaN |
| | **EWC 1, 3** | **2192** | **3901** | **2325** |
| | EWC 2, 1 | −2269 | −1915 | −2490 |
| | EWC 2, 2 | −3.12e4 | −3.23e4 | −1.15e5 |
| | EWC 2, 3 | −8.98e4 | −9.65e4 | −1.59e5 |
| W_BP | EWC 1, 1 | 1237 | 3055 | 1382 |
| | EWC 1, 2 | 1148 | 2800 | 1306 |
| | **EWC 1, 3** | **744** | **2000** | **−128** |
| | EWC 2, 1 | NaN | NaN | NaN |
| | EWC 2, 2 | 1027 | 3687 | 1416 |
| | EWC 2, 3 | NaN | NaN | NaN |