[Reviews · NeurIPS 2020]

Review 1

Summary and Contributions: This paper proposes a method, LPG-FTW, for lifelong multi-task reinforcement learning using factored policies that are comprised of a knowledge base shared across all tasks, L, and a set of task-specific parameters, s_t. On encountering a new task, first the task specific parameters are optimized with a policy gradient algorithm such that Ls_t yields a good policy for the new task. Subsequently, L is optimised to incorporate knowledge about this new policy while preserving knowledge about previous tasks, by penalising changes to previous policies in directions prescribed by their stored Hessians from the last time they were trained on. The method is an improvement on an existing method called PG-ELLA, which trains each new task on an entirely new policy before distilling the information into L, thereby forgoing any potential transfer from old tasks to new tasks. Theoretical guarantees are provided that show that LPG-FTW converges to the multi-task objective under certain conditions. The method is evaluated on sequences of continuous control tasks in Mujoco environments, where environment factors are randomly altered to generate new tasks, and on the MetaWorld10/50 tasks, and is shown to improve on a number of baselines across various lifelong learning metrics.

Strengths: The paper provides a novel method that simultaneously addresses two important challenges in lifelong learning: (i) how to use previously acquired knowledge to speed up new learning, (ii) how to update the agent’s knowledge without interfering with what it already knows. The method makes theoretical and empirical improvements on an existing related approach, PG-ELLA, with respect to both these challenges. Significant efforts are made to ensure that the experimental comparisons to other methods are fair, for example the full Hessian is used for the EWC method (a more powerful version than in the original paper, which only stores the diagonal approximation) and many other variants are also evaluated. The presentation of the results is excellent - the charts in Figures 2 and bottom of Figure 3 provide the information required to assess the balance of new learning and retention of old knowledge in each agent clearly.

Weaknesses: - One concern I have is how this method would perform, theoretically, in highly nonstationary environments. One of the assumptions for the theoretical guarantees is that the Hessians and policy gradients at the end of each task are iid - it’s not clear to me that this is a fair assumption for general lifelong learning, where one of the key conditions is that the data stream is not iid. - On a related note but from an empirical standpoint, it is also not clear how much nonstationarity there is in the sequence of tasks used for evaluation. In the first set of experiments, the environmental factors (like gravity, body part sizes) are chosen from a uniform random distribution, which resembles more of a traditional interleaved multi-task setting than a continual/lifelong learning one, since tasks with “nearby” parameters will presumably be very similar, and thereby not imposing a great burden on the agent to remember for long periods of time. How would the method perform if you linearly increased gravity or body part sizes over time for example? The ordering of tasks for MT10/50 is not clear either - are similar tasks grouped together or are they sampled from at random? Indeed it is acknowledged in the paper that one potential drawback of the method is that policies are only searched for in the span of L_t-1; could this be a real problem in a truly nonstationary environment if the method overfits to earlier parts of the task distribution, precluding it from being able to represent good policies from a different parts of the task distribution encountered later on during training? The paper does run experiments to demonstrate the diversity of tasks trained on (Appendix C), but this does not automatically induce nonstationarity over long timescales. - One of the common constraints imposed on lifelong/continual learning agents is that the memory capacity of the agent is limited, or that it grows slowly. LPG-FTW has the disadvantage that it must store the Hessian for each new task, which could take up a lot of memory for large models.

Correctness: The theoretical claims are justified, under the stated assumptions, and the empirical methodology seems correct.

Clarity: The paper is very clear and well written.

Relation to Prior Work: Yes.

Reproducibility: Yes

Additional Feedback: *** POST-REBUTTAL COMMENTS *** Thank you to the authors for their response. I appreciate the extra experiments run in a more nonstationary setting. I do believe that the paper will benefit *greatly* by making it much clearer that it is designed for an *easier* version of the general lifelong learning problem, i.e. one in which the tasks are sampled iid. It is a key assumption for the theoretical guarantee, and the extra experiments shown in a more nonstationary setting show that empirically it does not always improve on baselines. I think this method is a valuable contribution to the field, but I urge the authors to clarify the setting it is applicable for. ****** - How sensitive is the performance of the method to hyperparameters k, lambda and mu? - Given that the evaluation may be closer to the multi-task setting, is it possible that the problem that LPG-FTW is solving (and results in its superior performance in the evaluations) is closer to the one tackled in this paper [1] ? There the argument is made that the gradients of different tasks conflict with each other and so it can be hard to learn even in the multi-task setting, where the data stream is iid, especially when there is high curvature (perhaps the way LPG-FTW constrains the parameters along the Hessians helps with this issue?). - Was any backward transfer observed? I.e. did training on new related tasks improve the performance of older ones? - The ER baseline seems particularly bad for the MetaWorld experiments. In [2], multi-headed SAC gets an 85% success rate on MT10 in a multitask setting; of course a different RL algorithm is used, which may explain the difference, but given that ER should further shift the training paradigm to a multitask setting from a lifelong learning setting due to interleaved training on past data, it seems surprising to me how badly it performs. Perhaps some intuitions for this could be provided. - Overall, I think the paper could be strengthened if the degree of nonstationarity in the experiments were discussed in more detail or if results were shown in more nonstationary settings. [1] Yu, Tianhe, et al. "Gradient surgery for multi-task learning." arXiv preprint arXiv:2001.06782 (2020). [2] Yu, Tianhe, et al. "Meta-world: A benchmark and evaluation for multi-task and meta reinforcement learning." Conference on Robot Learning. 2020.


Review 2

Summary and Contributions: The paper presents a factored policy representation to train via policy gradients {PG} for lifelong learning to enable forward transfer but prevent catastrophic forgetting. The formulation introduces a set of learnable weights for every task which are optimised to construct the policy from a set of entries in a kind of policy parameter dictionary. These weights are updated via a PG algorithm while the dictionary is updated by an approximate multi task objective. The evaluation is performed on variations of OpenAI gym tasks and the multi-task (10/50) domains from Metaworld. Performance varies and it outperforms baselines across some tasks in the gym domains but consistently in Metaworld. The paper points in a valuable direction but is missing when it comes to connection to the existing field of continual/lifelong learning. The choice of individual baselines to present complete areas is not sufficient and the reasoning underlying the 'single-model assumption' remains sadly unsupported (the whole dictionary is also just a single model that is differently factorised). I appreciate some clarifications and the incremental update of a single dictionary. That simply higher capacity for EWC does not suffice for similarly addressing catastrophic forgetting is a valuable ablation and I hope this contributes (together with more quantitative information about the changes) to improve the paper. Statements like 'EWC suffers from catastrophic forgetting, unlike our approach.' do miss additional experimental support and it is likely that the approach would similarly suffer given more tasks. It would be great to see when this happens in terms of how many tasks are possible without the effect. In general both EWC and the presented method should be negatively affected when the tasks are not IID and the reasoning for this aspect remains unclear. Overall, a future iteration of the paper could be a valuable contribution but I am of the opinion that the current version does not meet the standards for NeurIPS and requires substantial improvements.

Strengths: The factored policy representation with a dictionary not of skills (as complete policies) but parameter combinations provides an interesting perspective on how information can be shared across tasks (in addition to e.g. mixture and product policies).

Weaknesses: One of the principal limitations is the a very restricted set of baselines. There have been strong methods before and after EWC and PG-ELLA including e.g. progressive nets (cited in the paper), progress and compress [1]. The method section remains unclear about previous work and novel contributions. After reading the PG-ELLA paper, it seems like a considerable part of the method section is the original method and it has to be made clear where exactly it is adapted. Statements such as line 129 will need to be explained in more detail to enable to reader to evaluate novel contributions. The current version is also missing a conclusion section to summarise the findings which would considerable help to shape the paper. [1]Schwarz, Jonathan, et al. "Progress & compress: A scalable framework for continual learning." arXiv preprint arXiv:1805.06370 (2018).

Correctness: In the original PG-ELLA paper [3], the method is compared against a PG single task baseline, such that it comes as a surprise that the submission equates single task learning and PG-ELLA. The reasoning remains unclear and has to be explicitly justified.

Clarity: Overall the paper is well written and clear. It is a bit heavy in notation which on its own is not a problem but (if I’m not mistaken) there seem to be a couple of inconsistent notations. E.g. vec(L) vs vec(L_t) (in text and algorithm); A might have to be A_t. Some other aspects of notation and symbols are introduced without sufficient explanation: usage of t^ and t, alpha, A, lambda. Additionally it seems as if we actually are required to keep one dictionary for every task (to compute the loss for Equation 2 over all tasks) - and I might be wrong here as the writing is a bit ambiguous - independent if this is the case, it would require some clarity (including the potential additional memory consumption).

Relation to Prior Work: As mentioned earlier, this seems one of the main weaknesses as the connection between PG-ELLA (and previous extensions) and this work remains ambiguous in the main paper. Statements such as given in line 129 should be extended to clarify the differences.

Reproducibility: No

Additional Feedback: Major points: -The introduction talks about limitations of different forms of lifelong learning. This requires references. -Line 96: ’without exploiting information from prior tasks’ is only accurate for non-shared policies as otherwise the policy includes information distilled from other tasks. This should be emphasised similar to the argument in the introduction. -Line 53: ‘all tasks must be very similar, or the model must be over-parameterized’ this applies to any policy form. Either the tasks must be more similar or we must introduce higher capacity models. It would help to be more concise here. -REINFORCE seems to be not used and directly replaced by NPG - best to skip the section if that is the case. -How do you count for additional experience in initialising L (this seems to be a separate training process) - this has to be added to the data used by LPG-FTW if not done already. -Why is single task = PG ELLA in 5.1 but not 5.2? This seems quite confusing as both domains are multitask. -How do you ensure comparable capacity across approaches? this might be a considerable disadvantage for eg EWG if the model is the same size as a single STL -How does training progress through a task sequence in Sec 5.1-5.2? After how many iterations do you switch and do you ensure to not train twice on the same task? Minor points: -‘Explore the policy space’ statement remains unclear. What does in mean here? -Regarding related work: there are many other approaches that also use experience from prior tasks when training on the next in a lifelong learning setting [1],[2] and many others. -It would help to annotate the algorithm. (Personal comment:) I find the factored policy representation quite interesting, which already was introduced with PG-ELLA. It might be interesting to look into the connection of GPI and GPE as factored value functions [4]. -Line 259: what does fixed number of iterations mean -> how many? -Line 136 ‘fundamentally different from prior methods that learn each task’s parameter vector in isolation’ could be misunderstood such that all prior method approach the problem in this way. Better to be more concise. -How can you evaluate EWC and others after algorithmic steps that only appear in PG-ELLA? (Sec 5.1) -Why does EWC performance drop after training on all tasks? (Figure 2) -Why do some algorithms have higher performance at start of training - without data? (Figure 1) -The function of the broader impact section for ethical purposes has been misunderstood but that is easy to fix.


Review 3

Summary and Contributions: A method is presented to enable life-long learning - training on a sequence of tasks without forgetting previous ones. The approach is similar to the PG-ELLA one (Ammar et al., 2014), where the policy on any given task is factorized into separate ‘skills’, but differs in the loss used to obtain that factorization: the requirement that the skills matrix is updated around an optimum for the new task is relaxed by adding terms linear in the gradient to the objectives. The authors include several experiments comparing their method to PG-ELLA and EWC, showing significantly stronger performance on catastrophic forgetting scores. Theoretical analysis is provided, indicating that while computationally expensive, the method converges to the optimal multi-task objective.

Strengths: The innovation presented in the paper is intuitively appealing: when trying to master many tasks, being overly focused on being optimal at the current one is not a good idea. The results are convincing on catastrophic forgetting, and the baselines are well-established.

Weaknesses: As with PG-ELLA, the computations required are expensive, as they involve Hessians. The connections to PG-ELLA are not discussed as extensively as warranted by the similarities. The innovation is also not discussed as extensively as warranted: how to let an agent balance different priorities is a topic that generally gets less attention than deserved, and this paper proposes an interesting method for doing just that.

Correctness: The method is compared to relevant baselines, on established task suites. I did not check the mathematical claims in detail, but they are very similar to those presented in the PG-ELLA paper and look plausible, so I believe they are correct.

Clarity: The paper is generally well-written. The mathematics is a bit tedious to get through, section 4 would benefit from having fewer symbols and more explanation of the method. For example, the authors could point out more explicitly how they modify the existing PG-ELLA algorithm by the addition of terms proportional to the gradient, and provide an intuitive meaning of those terms.

Relation to Prior Work: The paper is closely related to but at the same time importantly different from an existing method, which is cited and used as a baseline. The difference is discussed, although this could be made more explicit.

Reproducibility: Yes

Additional Feedback: My main suggestion is probably fairly clear from the above. Two more question/suggestions: 1. It is not completely clear to me how to connect the average task performance curves to the bar charts - shouldn't the end of the learning curve correspond to the 'final' bar? 2. Why does the STL do so well? Is it possible that the tasks are too similar after all? I would welcome a bit more discussion here.

[Author Response · NeurIPS 2020]

Thank you for noting the novelty of LPG-FTW, the fairness of evaluation, and the presentation and strength of results.

**R1**: Thank you for the very positive comments and detailed suggestions (**requested non-stationary results below**).

**3) I.i.d. assumption and non-stationarity**: We assume tasks are drawn i.i.d. from a task distribution. In our evaluation,
we sample tasks at random, and vary their order across trials (line 249). LPG-FTW supports interleaved MTL, but we
evaluate a true lifelong setting, without revisiting tasks. Following your suggestion, we ran LPG-FTW on domains
violating this i.i.d. assumption, with gravity increasing linearly in $[0.5, 1.5]g$. LPG-FTW, although not specifically
designed to handle non-stationarity, performs equivalently to the i.i.d. setting on Half-Cheetah & Hopper (Fig. 1). We
did no parameter tuning due to time; we think Walker2D needs tuning to handle the violated assumption.
**3) Storing all Hessians**: We don't store $H$'s in practice after updating $A, b$, since LPG-FTW only needs $H$ if revisiting
tasks, which we don't do. If tasks are revisited (e.g., interleaved MTL), it would require storing $H$ at a cost of $O(d^2T)$.
**8) Poor performance of experience replay (ER)**: as stated in Sec. 2, ER in PG algorithms (via importance sampling) is
unstable. If the policy moves far from earlier tasks, replay stops helping, which is aggravated as training progresses.

**R3**: Thank you. We address as many comments as space allows, and will update our draft with cites and clarifications.
**3&6) Previous work**: LPG-FTW improves PG-ELLA in 3 main ways. 1) PG-ELLA trains single-task (STL) policies
on each task separately and then finds $L, s^{(t)}$ via dictionary learning on the STL policies (line 60), while LPG-FTW
learns directly in the factored space (finding $s^{(t)}$ given the current $L$). 2) PG-ELLA has no explicit initialization, while
LPG-FTW uses Algo. 2, ensuring that columns of $L$ are diverse. 3) PG-ELLA assumes STL finds optimal policies,
while LPG-FTW adds a linear term to the cost to address non-optimality (line 129). This enables LPG-FTW to work on
far more complex tasks than PG-ELLA (Meta-World vs cartpole). We will emphasize these connections in Sec. 4.
**3) Chosen baselines**: We chose EWC and PG-ELLA as representatives of single- & multi-model classes. Progressive nets
scale poorly in lifelong settings. P&C suffers from the same limitations as EWC due to the single-model assumption,
and so it wouldn't scale well to the highly diverse tasks we study in Meta-World.
**4) Correctness of PG-ELLA evaluation**: We consulted with one author of PG-ELLA to validate that our evaluation was
correct. In PG-ELLA, the lifelong-learned policies are used only as a warm start for subsequent learning. They first
train STL policies, then run dictionary learning on the pre-trained policies to find an $L$ matrix, and finally use the $Ls^{(t)}$
policies to start a second STL process—for evaluation. Their main result (Fig. 2 in PG-ELLA) shows that initializing
STL from a lifelong-learned policy accelerates training. In contrast, our evaluation is entirely lifelong learning: the
agents are evaluated as they train on each task sequentially. There, PG-ELLA *does not* leverage information across tasks
(Fig. 1 and 3-top in our paper). Once all tasks are trained via STL, PG-ELLA runs dictionary learning and we evaluate
the policy in bars 3 and 4 of Fig. 2 and 3-bottom. Bar 4 corresponds to the starting point of the evaluation in PG-ELLA.
**5) Clarity of notation**: LPG-FTW keeps one dictionary shared across tasks: the multi-task cost in Eq. 2 is optimized
incrementally with auxiliary matrices $A, b$. $L_t$ denotes the $L$ learned up to task $t$. We only keep one $A$ matrix for all
tasks ($A_t$ is an auxiliary variable). $\hat{t}$ indexes over previous tasks, and $t$ is the current task. $\alpha$ is the policy obtained for a
task after its training process (using the $L$ *up to the previous task*). We will emphasize these clarifications in the text.
**8) Additional experience initializing $L$?**: There is none: initialization replaces the standard training for tasks $1$–$k$.
**8) Why is STL=PG-ELLA in 5.1 but not 5.2?**: Figs. 1 and 3-top show the training process, for which PG-ELLA uses
STL. Figs. 2 and 3-bottom show subsequent steps (bars 3 and 4) where PG-ELLA combines information from all tasks.
**8) Comparable capacity?**: All methods used policies of the same size, but overall capacity naturally varied across
methods. We followed your suggestion of increasing the capacity of EWC to match that of LPG-FTW in Meta-World,
and found that results didn't change significantly (Fig. 2).
**7) Reproducibility**: The clarifications above, along with the full code we provided, should make our results reproducible.
**8) How does training progress?**: Number of iterations is summarized Appendix B. Each task is seen by the agent only
once, and there is no possibility of going back for further experience at any point during the process.
**8) EWC performance drop after training all tasks**: EWC suffers from catastrophic forgetting, unlike our approach.

**R4**: Thank you for the very positive comments. For connections to PG-ELLA, see **R3**.
**8.1) Connections between charts**: 'Tune' matches the end of the curves, considering only per-task training but not how
that affects previous tasks. 'Update' and 'Final' come later: 'Final' assesses performance after all tasks are trained.
**8.2) Does STL performance imply tasks are similar?**: STL *ignores* all other tasks when training on one task, so task
similarity does not play a role. STL performs well because STL performance was used in the design of the benchmarks.

Fig. 1: Non-stationary results requested by **R1**. STL performance (for reference) measured on original i.i.d. task distribution.

Fig. 2: Higher EWC capacity, requested by **R3**.

[Meta-Review · NeurIPS 2020]

The reviewers were split on this paper, with two indicating accept and one indicating reject. The core contribution is to improve current methods for lifelong multi-task reinforcement leaning by transferring knowledge between tasks. The proposed method provides a mechanism to limit interference between tasks, with theoretical and empirical validation. The reviews also found weaknesses, including (a) ambiguity about the algorithm's performance in a non-stationary setting, (b) the relationship with PG-ELLA, (c) comparisons to stronger baselines, and (d) overly broad claims. The author rebuttal clarified several points, with an experiment in a non-iid setting (a), and a better explanation of the relationship with PG-ELLA (b). The post rebuttal discussion also covered areas where the paper should be improved. These include a better discussion of the assumptions behind the work (stationarity is commonly assumed in the literature but is constraining in a lifelong learning setting), the chosen baselines (adequate but could be stronger), and the language (several claims should be stated more carefully, with references where appropriate). The remaining weaknesses are minor and should be corrected by the authors in the final version. I therefore recommend acceptance.